# Pathogenic variants of sphingomyelin synthase SMS2 disrupt lipid landscapes in the secretory pathway

Tolulope Sokoya[1†], Jan Parolek[1†], Mads Møller Foged[2], Dmytro I Danylchuk[3], Manuel Bozan[1], Bingshati Sarkar[1], Angelika Hilderink[1], Michael Philippi[4], Lorenzo D Botto[5], Paulien A Terhal[6], Outi Mäkitie[7], Jacob Piehler[4], Yeongho Kim[8], Christopher G Burd[8], Andrey S Klymchenko[3], Kenji Maeda[2], Joost CM Holthuis[1]*

[1]Molecular Cell Biology Division, Department of Biology and Center of Cellular Nanoanalytics, Osnabrück University, Osnabrück, Germany; [2]Cell Death and Metabolism Group, Center for Autophagy, Recycling and Disease, Danish Cancer Society Research Center, Copenhagen, Denmark; [3]Laboratoire de Bioimagerie et Pathologies, Université de Strasbourg, Strasbourg, France; [4]Biophysics Division, Department of Biology and Center of Cellular Nanoanalytics, Osnabrück University, Osnabrück, Germany; [5]Division of Medical Genetics, Department of Pediatrics, University of Utah, Salt Lake City, United States; [6]Department of Genetics, University Medical Center Utrecht, Utrecht, Netherlands; [7]Children's Hospital, University of Helsinki and Helsinki University Hospital, Helsinki, Finland; [8]Department of Cell Biology, Yale School of Medicine, New Haven, United States

*For correspondence:
jholthuis@uni-osnabrueck.de

†These authors contributed equally to this work

**Abstract** Sphingomyelin is a dominant sphingolipid in mammalian cells. Its production in the *trans*-Golgi traps cholesterol synthesized in the ER to promote formation of a sphingomyelin/sterol gradient along the secretory pathway. This gradient marks a fundamental transition in physical membrane properties that help specify organelle identify and function. We previously identified mutations in sphingomyelin synthase SMS2 that cause osteoporosis and skeletal dysplasia. Here, we show that SMS2 variants linked to the most severe bone phenotypes retain full enzymatic activity but fail to leave the ER owing to a defective autonomous ER export signal. Cells harboring pathogenic SMS2 variants accumulate sphingomyelin in the ER and display a disrupted transbilayer sphingomyelin asymmetry. These aberrant sphingomyelin distributions also occur in patient-derived fibroblasts and are accompanied by imbalances in cholesterol organization, glycerophospholipid profiles, and lipid order in the secretory pathway. We postulate that pathogenic SMS2 variants undermine the capacity of osteogenic cells to uphold nonrandom lipid distributions that are critical for their bone forming activity.

## Editor's evaluation

Sphingomyelin synthase 2 (SMS2) is an enzyme located in the Golgi apparatus and the plasma membrane (PM) that mediates the synthesis of sphingomyelin (SM), a critical lipid in the PM. Mutations in SMS2 underlie a rare genetic disorder of bone formation. This useful study shows that the disease mutations cause retention of SMS2 in the ER, producing SM in the wrong place and leading to a disrupted SM/cholesterol gradient in the membranes of the secretory pathway. In addition to highlighting the roles of lipid gradients in cellular signaling pathways, this study also provides cell biologists with new tools to examine lipid localization in cells.

## Introduction

Eukaryotic membranes consist of complex lipid mixtures, with amounts and ratios of the individual lipids showing marked variations between organelles and membrane leaflets (*van Meer et al., 2008*; *Harayama and Riezman, 2018*). Although some rare lipids contribute to organelle function through stereospecific recognition by lipid binding proteins (*Di Paolo and De Camilli, 2006*), numerous recognition processes on or within organellar bilayers are determined by biophysical membrane properties that result from the collective behavior of the bulk lipids. Particularly striking are the lipid-induced changes in bilayer-thickness, lipid packing density and surface charge that accompany the transition from early to late organelles in the secretory pathway (*Bigay and Antonny, 2012*; *Holthuis and Menon, 2014*; *Sharpe et al., 2010*). These changes are highly conserved across species and provide specific cues for membrane proteins that govern vital processes such as protein secretion and signaling (*Bigay and Antonny, 2012*; *Magdeleine et al., 2016*; *Zhou and Hancock, 2018*). To defend the unique lipid mixtures of secretory organelles against erosion by vesicular transport, cells exploit cytosolic transfer proteins that enable specific lipids to bypass vesicular connections by mediating their monomeric exchange at contact sites between distinct organelles (*Wong et al., 2019*). Moreover, organelles like the ER harbor membrane property sensors that provide feedback to the lipid metabolic network to preserve their characteristic lipid composition when exposed to stress or metabolic insults (*Levental et al., 2020*; *Radanović et al., 2018*).

Sterols and sphingomyelin (SM) are prime examples of bulk membrane lipids that are unevenly distributed between secretory organelles (*van Meer et al., 2008*). Sterols are rare in the ER but abundant in the *trans*-Golgi and plasma membrane (PM). The bulk of SM is synthesized in the lumen of the *trans*-Golgi from ceramides supplied by the ER and delivered by vesicular transport to the PM, where it accumulates in the exoplasmic leaflet (*Hanada et al., 2003*). SM is the preferred binding partner of cholesterol (*Slotte, 2013*). About one-third of the total cholesterol pool in the PM is sequestered by SM (*Das et al., 2014*; *Endapally et al., 2019*). Besides influencing cellular cholesterol homeostasis, SM contributes to an enhanced packing density and thickening of *trans*-Golgi and PM bilayers. This, in turn, may modulate protein sorting by hydrophobic mismatching of membrane spans (*Munro, 1995*; *Quiroga et al., 2013*). Moreover, an asymmetric distribution of SM across late secretory and endolysosomal bilayers is relevant for an optimal repair of damaged organelles. Lysosome wounding by chemicals or bacterial toxins triggers a rapid $Ca^{2+}$-activated scrambling and cytosolic exposure of SM (*Ellison et al., 2020*; *Niekamp et al., 2022*). Subsequent conversion of SM to ceramides by neutral SMases on the cytosolic surface of injured lysosomes promotes their repair, presumably by driving an inverse budding of the damaged membrane area in a process akin to ESCRT-mediated formation of intraluminal vesicles. This SM-based membrane restoration pathway functions independently of ESCRT and may also operate at the PM (*Niekamp et al., 2022*).

SM biosynthesis in mammals is mediated by SM synthase 1 (SMS1) and SMS2. Both enzymes act as phosphatidylcholine (PC):ceramide phosphocholine transferases, which catalyze the transfer of the phosphorylcholine head group from PC onto ceramide to generate SM and diacylglycerol (DAG) (*Huitema et al., 2004*). SMS1 resides in the *trans*-Golgi, and its deficiency in mice causes mitochondrial dysfunction and disrupts insulin secretion (*Yano et al., 2013*; *Yano et al., 2011*). SMS2 resides both in the *trans*-Golgi and at the PM. Its deficiency ameliorates diet-induced obesity and insulin resistance (*Kim et al., 2018*; *Li et al., 2011*; *Mitsutake et al., 2011*; *Sugimoto et al., 2016*). Removal of SMS1 or SMS2 has only a minor impact on ceramide, DAG and SM pools in tissues or cells, and the mechanisms underlying the phenotypes observed in SMS1 and SMS2 knockout mice are not well understood. Besides SMS1 and SMS2, mammalian cells contain an ER-resident and SMS-related protein (SMSr), which displays phospholipase C activity and synthesizes trace amounts of the SM analog ceramide phosphoethanolamine (*Murakami and Sakane, 2021*; *Vacaru et al., 2009*).

We previously reported that SMS2 is highly expressed in bone and identified heterozygous mutations in the SMS2-encoding gene (*SGMS2*) as the underlying cause of a clinically described autosomal dominant genetic disorder – osteoporosis with calvarial doughnut lesions (OP-CDL: OMIM #126550) (*Pekkinen et al., 2019*). The clinical presentations of OP-CDL range from childhood-onset osteoporosis with low bone mineral density, skeletal fragility and sclerotic doughnut-shaped lesions in the skull to a severe spondylometaphyseal dysplasia with neonatal fractures, long-bone deformities, and short stature. The milder phenotype is associated with the nonsense variant p.Arg50*, which gives rise to a truncated but catalytically active enzyme that mislocalizes to the *cis/medial*-Golgi (T. Sokoya

and J. Holthuis, unpublished data). However, the most severe phenotypes are associated with two closely localized missense variants, p.Ile62Ser and p.Met64Arg. Interestingly, these missense variants enhance de novo SM biosynthesis by blocking ER export of enzymatically active SMS2 (*Pekkinen et al., 2019*). This suggests that OP-CDL in patients with pathogenic SMS2 variants is not due to a reduced capacity to synthesize SM but rather a consequence of mistargeting bulk SM production to an early secretory organelle. How this affects the contrasting lipid landscapes and membrane properties in the secretory pathway remains to be established.

In this work, we used genetically engineered cell lines and OP-CDL patient-derived fibroblasts to address the impact of pathogenic SMS2 variants p.Ile62Ser and p.Met64Arg on the lipid composition, transbilayer arrangement, and packing density of early and late secretory organelles. Toward this end, we combined shotgun lipidomics on purified organelles with the application of lipid biosensors and targeted solvatochromic fluorescent probes in live cells. We show that cells harbouring pathogenic SMS2 variants accumulate PM-like amounts of SM in the ER and display a disrupted transbilayer SM asymmetry. This is accompanied by significant imbalances in cholesterol organization and membrane lipid order. We also find that pathogenic SMS2 variants cause marked changes in the ER glycerophospholipid profile, including an enhanced phospholipid desaturation and rise in cone-shaped ethanolamine-containing phospholipids, potentially reflecting an adaptive cellular response to counteract SM-mediated rigidification of the ER bilayer. Our data indicate that pathogenic SMS2 variants profoundly undermine the cellular capacity to uphold nonrandom lipid distributions in the secretory pathway that may be critical for the bone forming activity of osteogenic cells.

## Results

### The IXMP motif in SMS2 is part of an autonomous ER export signal

The most severe clinical presentations of OP-CDL are associated with the SMS2 missense variants p.I62S and p.M64R, which cause retention of a functional enzyme in the ER (*Pekkinen et al., 2019*). Ile62 and Met64 are part of a conserved sequence motif, IXMP, which is located 13–14 residues upstream of the first membrane span in both SMS1 and SMS2 (*Figure 1A, B*). We reasoned that this motif may be part of an ER export signal, which could explain its absence in the ER-resident SMS family member SMSr (*Vacaru et al., 2009*). To test this idea, we generated FLAG-tagged SMS2 constructs in which Ile62 or Met64 was replaced with Ser or Arg, respectively. Upon their transfection in HeLa cells, the subcellular distribution of the SMS2 variants was determined by fluorescence microscopy using antibodies against the FLAG tag and ER-resident protein calnexin. In agreement with our previous findings (*Pekkinen et al., 2019*), SMS2[I62S] and SMS2[M64R] were each retained in the ER, in contrast to wildtype SMS2, which localized to the Golgi and PM (*Figure 1C*; *Appendix 1—figure 1*). We then asked whether the IXMP motif in SMS2 can mediate ER export independently of other sorting information. To address this, we created a FLAG-tagged chimera protein, SMSr-SMS2[11-77], in which the region linking the *N*-terminal SAM domain and first membrane span of SMSr was replaced with the IXMP-containing cytosolic tail of SMS2 (*Figure 1A*). Contrary to SMSr, SMSr-SMS2[11-77] localized to the Golgi. However, SMSr-SMS2[11-77] variants in which Ile62 or Met64 was replaced with Ser or Arg, respectively, were retained in the ER (*Figure 1D*; *Appendix 1—figure 1*). This indicates that the IXMP motif in SMS2 is part of an autonomous ER export signal. Whether mutation of Ile62 or Met64 disrupts ER export by perturbing a linear sequence motif or the overall fold of the enzyme's *N*-terminal cytosolic tail remains to be established.

### Pathogenic SMS2 variants mediate bulk production of SM in the ER

Metabolic labeling of patient-derived fibroblasts with [14]C-choline showed that missense SMS2 variants p.I62S and p.M64R cause a marked increase in the rate of de novo SM biosynthesis (*Pekkinen et al., 2019*). To directly test the impact of these pathogenic mutations on the biosynthetic capacity of SMS2, we stably transduced SMS1/2 double knockout (ΔSMS1/2) HeLa cells with doxycycline-inducible expression constructs encoding FLAG-tagged SMS2, SMS2[I62S], SMS2[M64R] or their enzyme dead isoforms SMS2[D276A], SMS2[I62S/D276A], and SMS2[M64R/D276A], respectively. After treatment of cells with doxycycline for 16 hr, SMS2 expression was verified by immunoblot analysis and fluorescence microscopy (*Figure 2A, B*; *Figure 2—source data 1*; *Appendix 1—figure 2*). Next, control and doxycycline-treated cells were metabolically labeled with a clickable sphingosine analogue for 16 hr, subjected to

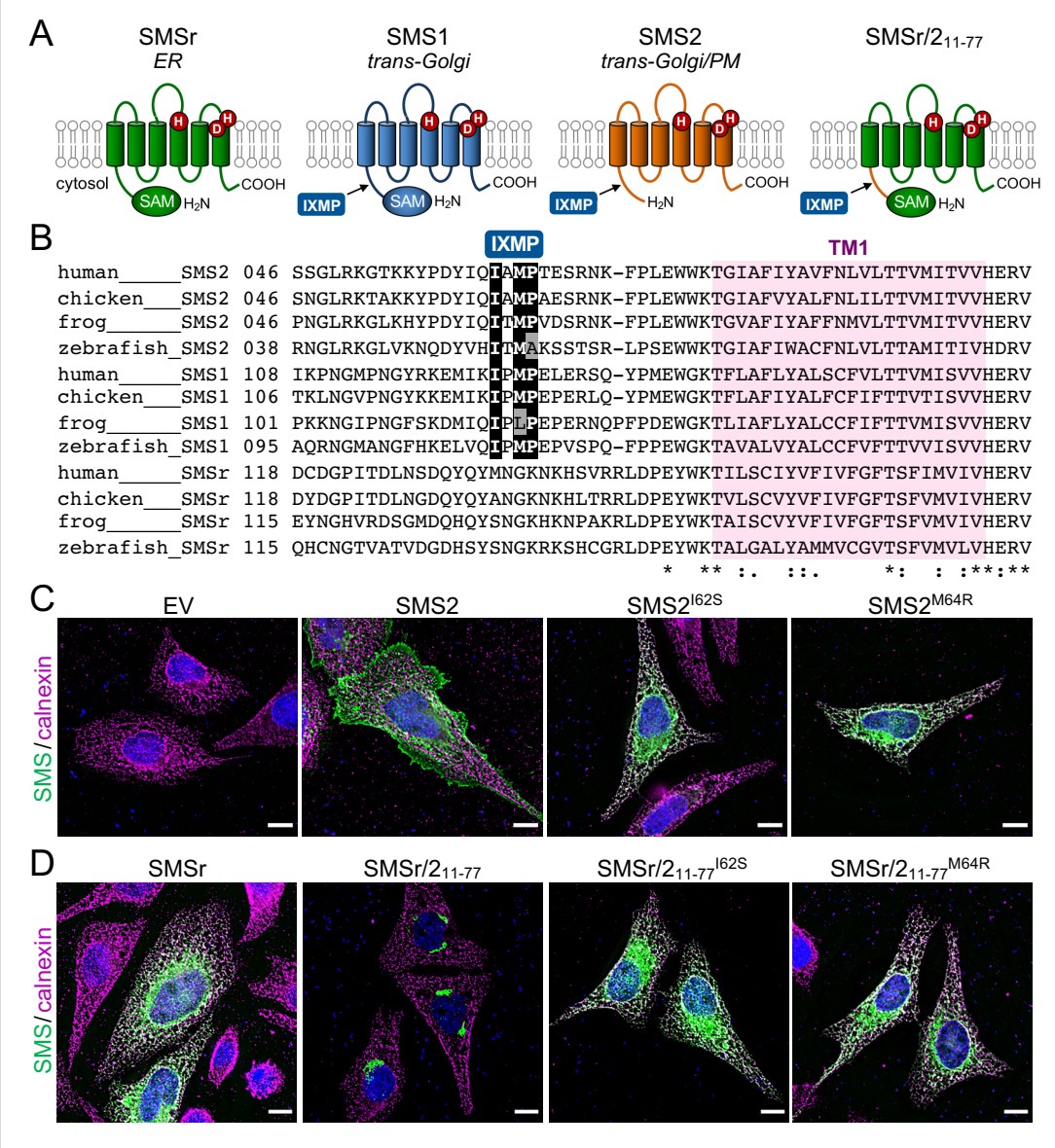

**Figure 1.** SMS2 contains an autonomous ER export signal. (**A**) Membrane topology of SMS family members and chimeric protein SMSr/2₁₁₋₇₇. Active site residues are marked in red. The position of a conserved IXMP sequence motif is marked by an arrow. (**B**) Sequence alignment of the region immediately upstream of the first membrane span (TM1) in vertebrate SMS family members. Note that human SMS2 residues Ile62 and Met64 are part of the IXMP sequence motif, which is conserved in SMS1 and SMS2, but not SMSr, across different vertebrate species. (**C**) HeLa cells transfected with empty vector (EV) or FLAG-tagged SMS2, SMS2$^{I62S}$, or SMS2$^{M64R}$ were fixed, immunostained with α-FLAG (*green*) and α-calnexin (*magenta*) antibodies, counterstained with DAPI (*blue*) and imaged by DeltaVision microscopy. (**D**) HeLa cells transfected with FLAG-tagged SMSr, SMSr/2₁₁₋₇₇, SMSr/2₁₁₋₇₇$^{I62S}$ or SMSr/2₁₁₋₇₇$^{M64R}$ were fixed, immunostained with α-FLAG (*green*) and α-calnexin (*magenta*) antibodies, counterstained with DAPI (*blue*) and imaged by DeltaVision microscopy. Scale bar, 10 µm.

total lipid extraction, click reacted with the fluorogenic dye 3-azido-7-hydroxycoumarin, and analyzed by TLC. This revealed that doxycycline-induced expression of SMS2$^{I62S}$ and SMS2$^{M64R}$, but not their enzyme-dead isoforms, fully restored SM biosynthesis in ΔSMS1/2 cells (*Figure 2C*).

Quantitative mass spectrometric analysis of total lipid extracts from wildtype and ΔSMS1/2 cells revealed that removal of SMS1 and SMS2 wiped out the entire cellular SM pool and caused a fourfold increase in glycosphingolipid (GSL) levels, consistent with a competition between Golgi-resident SM and glucosylceramide (GlcCer) synthases for ceramide substrate (*Figure 2D*; *Figure 2—source data 2*; *Appendix 1—figure 3A, B*; *Appendix 1—figure 3—source data 1*). In ΔSMS1/2 cells transduced with pathogenic SMS2$^{I62S}$ or SMS2$^{M64R}$, addition of doxycycline fully restored the SM pool. This was

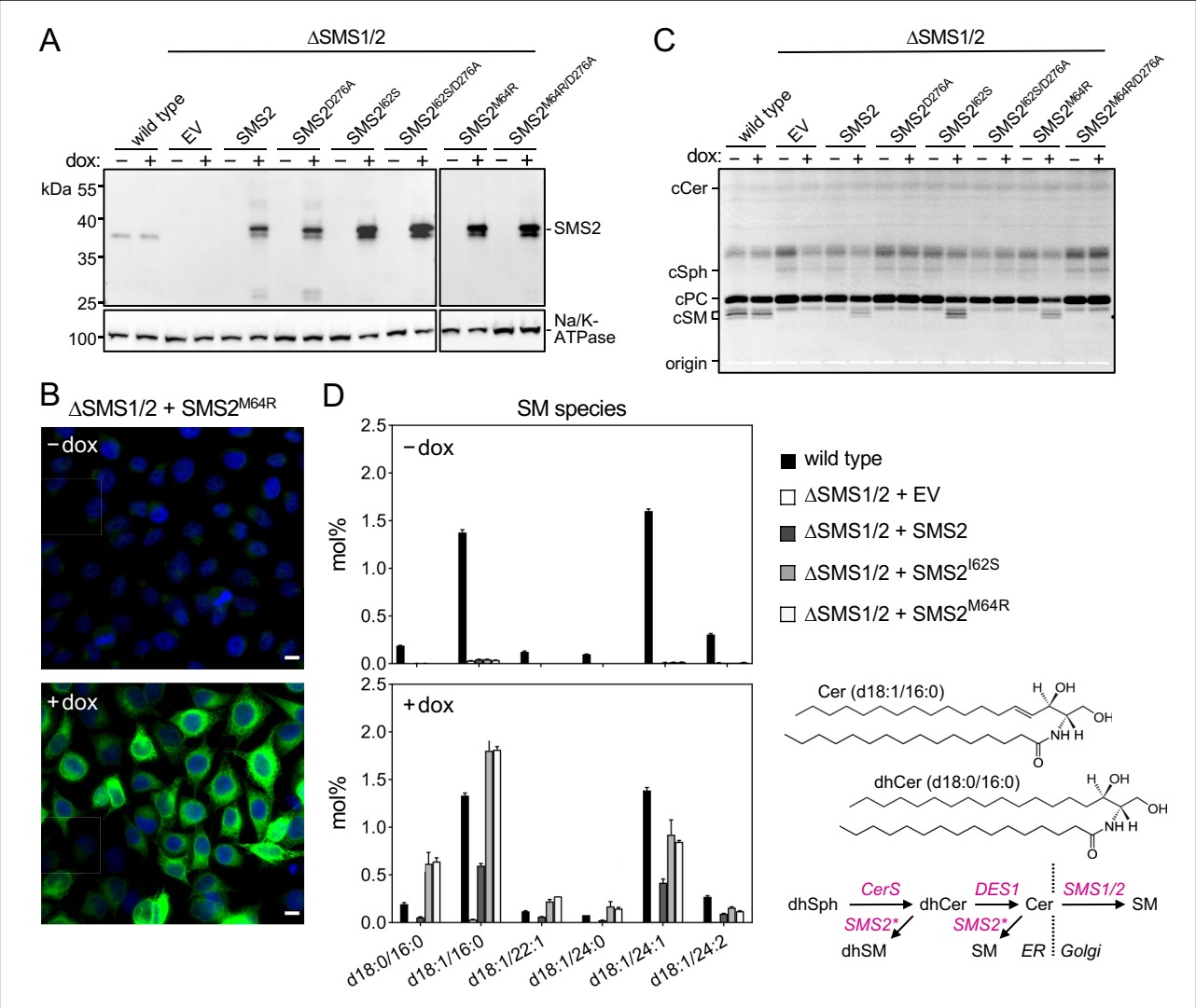

**Figure 2.** Pathogenic SMS2 variants support bulk production of SM in the ER. (**A**) HeLa SMS1/2 double KO (ΔSMS1/2) cells transduced with doxycycline-inducible constructs encoding FLAG-tagged SMS2, SMS2^I62S, SMS2^M64R or their enzyme-dead isoforms (D276A) were grown for 16 hr in the absence or presence of 1 μg/ml doxycycline and then subjected to immunoblot analysis using α-SMS2 and α-Na/K-ATPase antibodies. Wildtype HeLa cells served as control. (**B**) ΔSMS1/2 cells transduced with doxycycline-inducible FLAG-tagged SMS2^M64R were treated as in (**A**), fixed, immunostained with α-FLAG antibody (*green*), counterstained with DAPI (*blue*) and imaged by conventional fluorescence microscopy. Scale bar, 10 μm. (**C**) Cells treated as in (**A**) were metabolically labeled with a clickable sphingosine analogue for 16 h, subjected to total lipid extraction, click reacted with the fluorogenic dye 3-azido-7-hydroxycoumarin and analyzed by TLC. (**D**) SM species in total lipid extracts of cells treated as in (**A**) were quantified by LC-MS/MS and expressed as mol% of total phospholipid analyzed. Note that the rise in dihydroSM (dhSM, d18:0/16:0) in ΔSMS1/2 cells expressing SMS2^I62S or SMS2^M64R (SMS2*) is likely due to competition between ER-resident ceramide desaturase (DES1) and SMS2* for dihydroceramide (dhCer, d18:0/16:0), which is synthesized de novo by ceramide synthase (CerS) from dihydrosphingosine (dhSph). Data are average ± SD of three technical triplicates.

The online version of this article includes the following source data for figure 2:

**Source data 1.** Unprocessed and uncropped image files of the immunoblots.

**Source data 2.** Raw data of the quantitative analysis of SM species.

accompanied by a decrease in GSL levels. Doxycycline-induced expression of SMS2 only partially restored the SM pool, presumably because SMS2, unlike its pathogenic isoforms, has no direct access to ER-derived ceramides and must compete with GlcCer synthase for ceramides delivered to the Golgi. Moreover, ΔSMS1/2 cells expressing SMS2^I62S or SMS2^M64R contained three- to fourfold higher levels of dihydroceramide (Cer d18:0/16:0) and dihydroceramide-based SM (SM d18:0/16:0) than wildtype or SMS2-expressing ΔSMS1/2 cells (*Figure 2D*; *Figure 2—source data 2*; *Appendix 1—figure 3B*; *Appendix 1—figure 3—source data 1*), which suggests that ER-resident pathogenic SMS2

variants compete with ceramide desaturase DES1 for dihydroceramide substrate synthesized in the ER. All together, these data indicate that pathogenic SMS2 variants support bulk production of SM in the ER.

## Lipidome analysis of ER and PM isolated from cells expressing pathogenic SMS2 variants

We next asked whether pathogenic SMS2 variants that mediate bulk production of SM in the ER interfere with the ability of cells to generate a SM/cholesterol concentration gradient along the secretory pathway. To address this, we analyzed the lipid composition of ER and PM purified from wildtype or ΔSMS1/2 cells that express either SMS2 or the pathogenic variant SMS2^M64R. For purification of the ER, cells were lysed and a post-nuclear supernatant was incubated with an antibody against calnexin (*Figure 3A*). This was followed by incubation with secondary antibody-conjugated paramagnetic microbeads. For PM isolation, the surface of cells was treated with a non-membrane permeant biotinylation reagent before cell lysis (*Figure 4A*). A post-nuclear supernatant was then directly incubated with streptavidin-conjugated paramagnetic microbeads. The microbeads were applied to columns packed with ferromagnetic spheres (μMACS columns) and the bound material was eluted after the columns were thoroughly washed. The purity of isolated ER and PM was assessed by immunoblot and lipidome analysis.

As shown in *Figure 3B*; *Figure 3—source data 1*, ER eluates contained calnexin but were devoid of protein markers of the PM (Na/K-ATPase), lysosomes (LAMP1), and mitochondria (pMito60). As expected, ER eluates from cells expressing pathogenic variant SMS2^M64R contained readily detectable levels of the protein. In contrast, no traces of SMS2 were found in ER eluates from cells expressing the wildtype protein. As there is no specific lipid marker for the ER, using a lipidomics approach to confirm that pull-down with anti-calnexin antibody indeed isolates the ER is not trivial. However, the ER is known to synthesize ceramides whereas SM is primarily produced in the *trans*-Golgi and accumulates in the PM. In line with the immunoblot data, ER eluates from wildtype cells displayed a fivefold higher ceramide/SM ratio than total cell lysates (*Figure 3C*; *Figure 3—source data 2*). Moreover, ER eluates were largely devoid of lipids that are normally concentrated in mitochondria (cardiolipin, CL), PM (SM, cholesterol), and lysosomes (bis(monoacyl-glycerol)phosphate, here quantified together with the isobaric phosphatidylglycerol and reported as BMP/PG). Immunoblot analysis of the PM eluates revealed that they contain Na/K-ATPase but lack protein markers of the ER (calnexin), lysosomes (LAMP1), and mitochondria (pMito60; *Figure 4B*; *Figure 4—source data 1*). As expected, PM eluates from cells expressing wildtype SMS2 contained readily detectable amounts of the protein. On the other hand, PM eluates from ΔSMS1/2 cells expressing the pathogenic variant SMS2^M64R were devoid of this protein. Moreover, lipidome analysis of PM eluates revealed significantly elevated levels of lipids that are typically concentrated in the PM (i.e. SM, cholesterol, PS) and a fivefold reduction in the ceramide/SM ratio relative to total cell lysates (*Figure 4C*; *Figure 4—source data 2*). Lipids primarily associated with lysosomes and mitochondria (PG, CL) were largely absent.

## Cells expressing pathogenic SMS2 variants accumulate SM in the ER

Using the pull-down approaches described above, we next determined the lipid composition of the ER and PM isolated from wildtype and ΔSMS1/2 cells expressing SMS2 or SMS2^M64R. The ER from ΔSMS1/2 cells expressing SMS2 had a lipid composition similar to that of the ER from wildtype cells. In contrast, the ER from SMS2^M64R-expressing cells contained sevenfold higher SM levels, that is ~10 mol% SM instead of ~1.5 mol% of all identified lipids (*Figure 3D*; *Figure 3—source data 2*). This increase in ER-bound SM was accompanied by a twofold rise in DAG levels and a significant drop in the amount of PC and ceramide, consistent with the presence of a catalytically active SM synthase in the ER. Interestingly, expression of SMS2^M64R also led to a marked (1.8-fold) increase in ER-associated PE levels. In contrast, ER levels of cholesterol and other bulk lipids were largely unaffected. However, we noticed that expression of SMS2^M64R enhanced unsaturation of bulk phospholipid in the ER, as indicated by a significant rise in di-unsaturated PC at the expense of saturated and mono-unsaturated PC species (*Figure 3E*; *Figure 3—source data 2*). PC chain length, on the other hand, was largely unaffected. Strikingly, SMS2^M64R expression also caused a sharp increase in ER-bound ceramide-1-phosphate (Cer1P; *Figure 3D*; *Figure 3—source data 2*). Moreover, cellular Cer1P levels

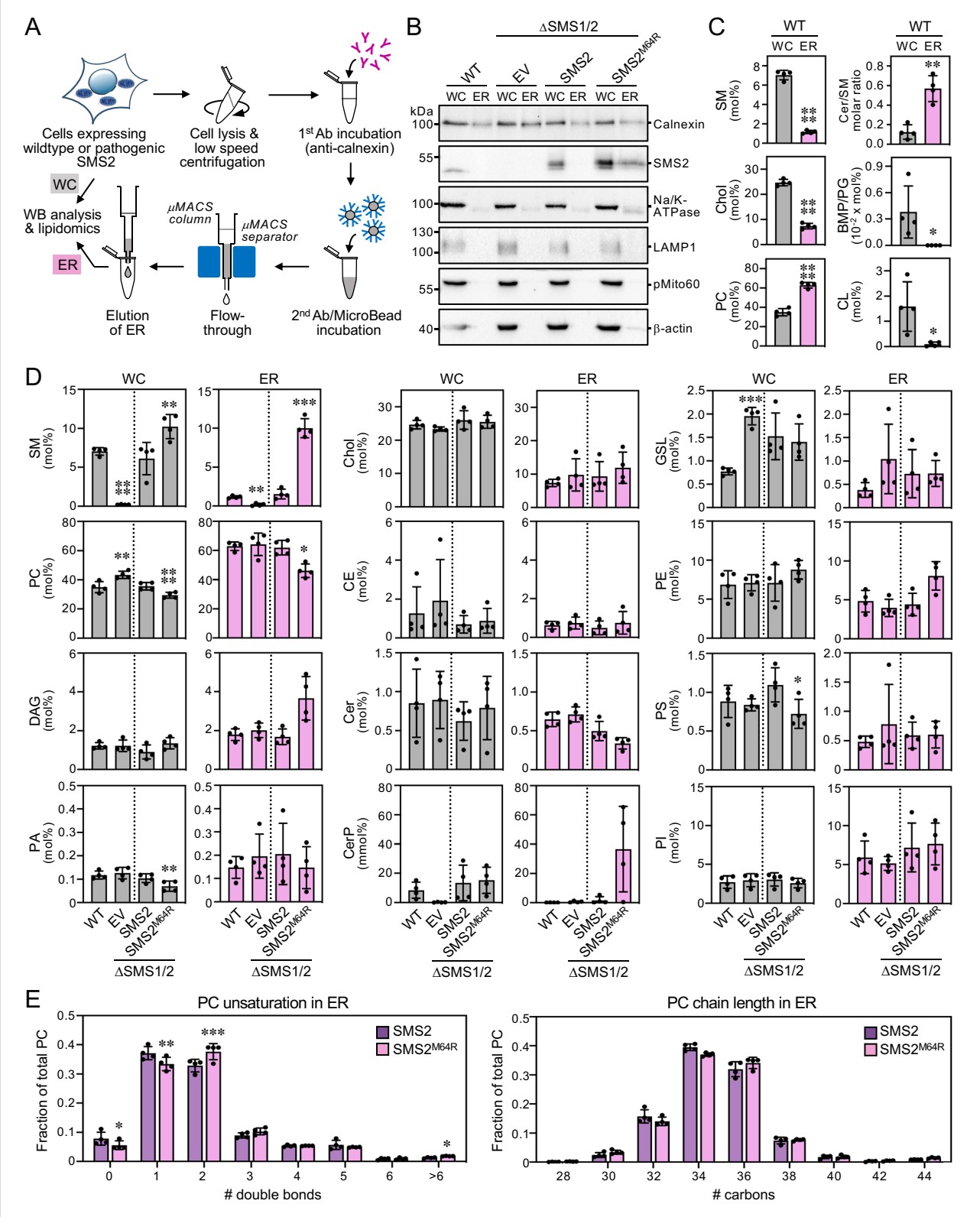

**Figure 3.** Cells expressing pathogenic variant SMS2^M64R accumulate SM in the ER. (**A**) Workflow for affinity purification of the ER from HeLa cells expressing wildtype or pathogenic SMS2 variants. (**B**) HeLa wildtype (WT) or ΔSMS1/2 cells transduced with empty vector (EV) or doxycycline-inducible SMS2 or SMS2^M64R were treated with doxycycline (1 µg/ml, 16 hr), lysed and used to purify the ER as in (**A**). Whole cell lysates (WC) and purified ER were subjected to immunoblot analysis using antibodies against SMS2 and various organellar markers. Samples were loaded on an equivalents basis.

*Figure 3 continued on next page*

Figure 3 continued

(**C**) Lipid composition of whole cell lysates (WC) and ER purified from HeLa wildtype cells (WT) was determined by mass spectrometry-based shotgun lipidomics. Levels of the different lipid classes are expressed as mol% of total identified lipids. CE, cholesteryl esters. (**D**) Lipid composition of whole cell lysates (WC) and ER purified from cells as in (**B**) was determined as in (**C**). (**E**) Comparative analysis of PC unsaturation and chain length in ER purified from ΔSMS1/2 cells expressing SMS2 or SMS2$^{M64R}$. The graphs show total numbers of double bonds (*left*) or carbon atoms (*right*) in the two acyl chains. All data are average ± SD, n=4. *p<0.05, **p<0.01, ***p<0.001, ****p<0.0001 by paired *t* test.

The online version of this article includes the following source data for figure 3:

**Source data 1.** Unprocessed and uncropped image files of the immunoblots.

**Source data 2.** Raw data of the quantitative analysis of lipid species in whole cells or isolated ER.

were essentially abolished in SM synthase-deficient cells, indicating that production of Cer1P is tightly coupled to SM biosynthesis.

The PM from ΔSMS1/2 cells expressing SMS2 had a SM content similar to the PM from wildtype cells (~10 mol%). In comparison, the PM from ΔSMS1/2 cells expressing SMS2$^{M64R}$ had a slightly reduced SM content (~8 mol%) even though the total SM content of these cells was considerably higher (*Figure 4D*; *Figure 4—source data 2*). PM-associated levels of cholesterol and other bulk lipids did not show any major differences among the various cell lines, except for a lack of SM and elevated PC and GSL content in SMS-deficient cells. The PM from all four cell lines contained significantly elevated levels of saturated PC species in comparison to the ER. In addition, the PM from ΔSMS1/2 cells expressing SMS2$^{M64R}$ contained fourfold higher levels of dihydroSM (represented by SM species with 0 double bonds; *Figure 4E*; *Figure 4—source data 2*), consistent with the ER residency of this enzyme. Collectively, these data indicate that pathogenic SMS2 variants disrupt the SM gradient along the secretory pathway and cause substantial changes in the lipid profile of the ER.

To confirm that cells expressing pathogenic SMS2 variants accumulate SM in the ER, we next used an engineered GFP-tagged version of equinatoxin II (Eqt) as non-toxic SM reporter in live cells. To enable detection of SM inside the secretory pathway, the reporter was equipped with the *N*-terminal signal sequence of human growth hormone and tagged at its *C*-terminus with oxGFP, yielding EqtSM$_{SS}$ (*Deng et al., 2016*). A luminal Eqt mutant defective in SM binding, EqtSol$_{SS}$, served as control. When expressed in human osteosarcoma U2OS cells, both EqtSM$_{SS}$ and EqtSol$_{SS}$ showed a reticular distribution that overlapped extensively with the ER marker protein VAPA (*Figure 5A*). However, upon co-expression with SMS2$^{M64R}$, EqtSM$_{SS}$ but not EqtSol$_{SS}$ displayed a distinct punctate distribution that coincided with the ER network. EqtSM$_{SS}$-containing puncta were not observed in cells expressing the enzyme-dead variant SMS2$^{M64R/D276A}$, indicating that their formation strictly relies on SM production in the ER (*Figure 5A*). To verify that the EqtSM$_{SS}$-positive puncta mark ER-resident pools of SM, U2OS cells co-expressing EqtSM$_{SS}$ and SMS2$^{M64R}$ were subjected to hypotonic swelling as described before (*King et al., 2020*). After incubation for 5 min in hypotonic medium, the ER's fine tubular network transformed into numerous micrometer-sized vesicles. In SMS2$^{M64R}$-expressing cells, the membranes of these ER-derived vesicles were extensively labelled with EqtSM$_{SS}$ (*Figure 5B*). In contrast, in hypotonic cells co-expressing EqtSM$_{SS}$ with SMS2$^{M64R/D276A}$ or EqtSol$_{SS}$ with SMS2$^{M64R}$, the reporter was found exclusively in the lumen of ER-derived vesicles, indicating that Eqt staining of the ER membrane critically relies on catalytically active SMS2$^{M64R}$ and a SM-binding competent reporter. In agreement with the ER lipidome analyses, these results demonstrate that cells expressing SMS2$^{M64R}$ accumulate bulk amounts of SM in the ER. Moreover, our finding that hypotonic swelling of SMS2$^{M64R}$-expressing cells transforms the ER-associated punctate distribution of EqtSM$_{SS}$ to a more uniform labeling of the ER bilayer suggests that alterations in membrane curvature and/or lipid packing may affect the lateral organization of Eqt-SM assemblies.

## Pathogenic SMS2 variants break transbilayer SM asymmetry

SM adopts an asymmetric distribution across the bilayers of late secretory and endolysosomal organelles, with the bulk of SM residing in the luminal/exoplasmic leaflet. However, using GFP-tagged EqtSM as cytosolic SM reporter (EqtSM$_{cyto}$), we found that perturbation of lysosome or PM integrity by pore-forming chemicals or toxins disrupts SM asymmetry by triggering a rapid transbilayer movement of SM catalyzed by Ca$^{2+}$-activated scramblases (*Niekamp et al., 2022*). To perform its central task in membrane biogenesis, the ER harbors constitutively active scramblases that enable a rapid equilibration of newly synthesized phospholipids across its bilayer (*Pomorski and Menon, 2016*). We therefore

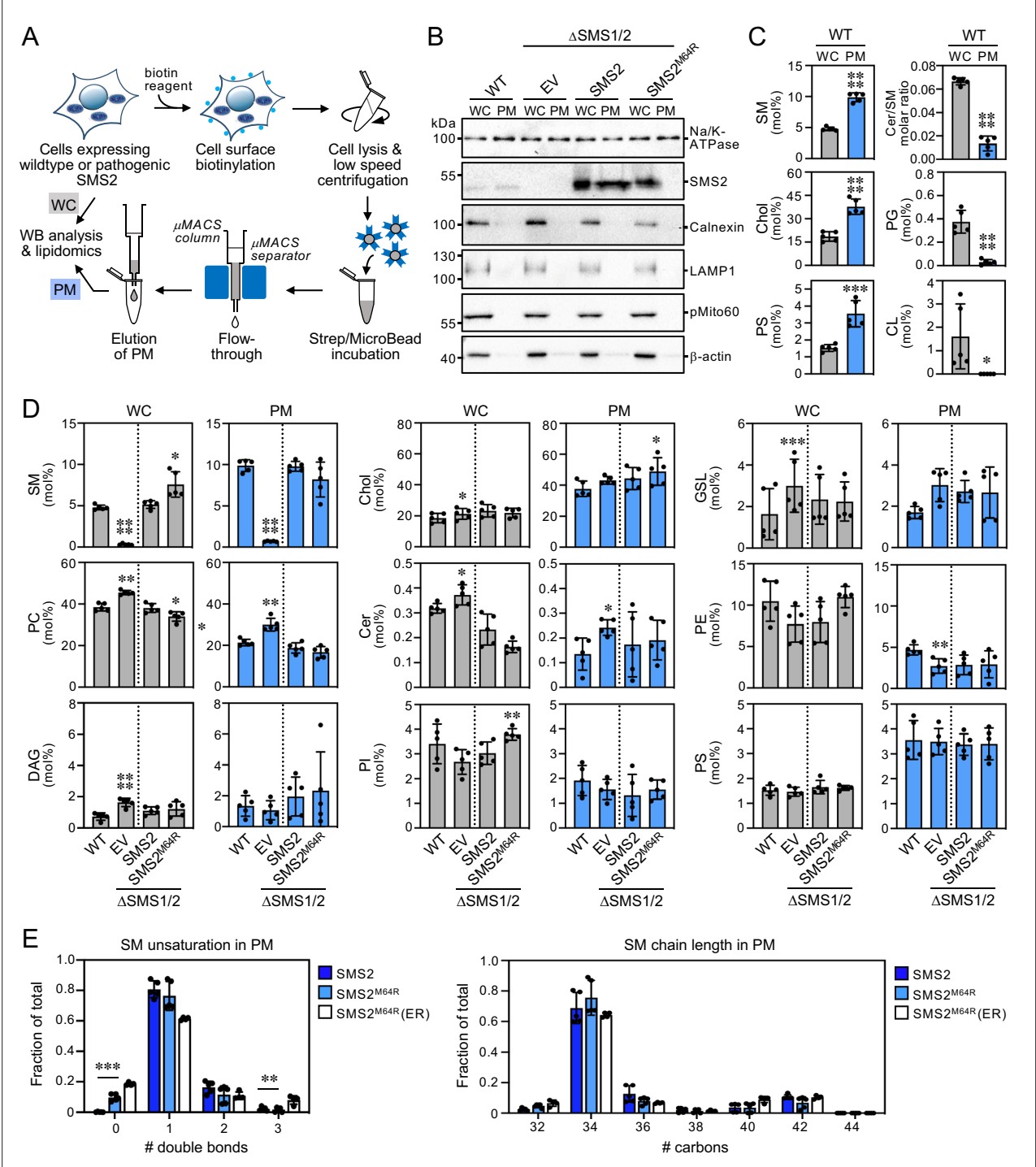

**Figure 4.** Lipid composition of the PM of cells expressing wildtype or pathogenic SMS2 variants. (**A**) Workflow for affinity purification of the PM from HeLa cells expressing wildtype or pathogenic SMS2 variants. (**B**) HeLa wildtype (WT) or ΔSMS1/2 cells transduced with empty vector (EV) or doxycycline-inducible SMS2 or SMS2^{M64R} were treated with doxycycline (1 µg/ml, 16 h), lysed and used to purify the PM as in (**A**). Whole cell lysates (WC) and purified PM were subjected to immunoblot analysis using antibodies against SMS2 and various organellar markers. Samples were loaded on an equivalents basis. (**C**) Lipid composition of whole cell lysates (WC) and PM purified from HeLa wildtype cells (WT) was determined by mass spectrometry-based shotgun lipidomics. Levels of the different lipid classes are expressed as mol% of total identified lipids. (**D**) Lipid composition of whole cell lysates (WC) and PM purified from cells as in (**B**) was determined as in (**C**). All data are average ± SD, n=5. (**E**) Comparative analysis of SM unsaturation and chain length in PM purified from ΔSMS1/2 cells expressing SMS2 or SMS2^{M64R}. Data on SM unsaturation and chain length in ER purified from ΔSMS1/2 cells

*Figure 4 continued on next page*

*Figure 4 continued*

expressing SMS2$^{M64R}$ are included as reference. The graphs show total numbers of double bonds (*left*) or carbon atoms (*right*) in the sphingoid base and acyl chain. All data are average ± SD, n=5. *p<0.05, **p<0.01, ***p<0.001, ****p<0.0001 by paired *t* test.

The online version of this article includes the following source data for figure 4:

**Source data 1.** Unprocessed and uncropped image files of the immunoblots.

**Source data 2.** Raw data of the quantitative analysis of lipid species in whole cells or isolated PM.

asked whether SM produced by pathogenic SMS2 variants in the ER lumen has access to the cytosolic leaflet. As expected, EqtSM$_{cyto}$ in wildtype or ΔSMS1/2 cells expressing SMS2 displayed a cytosolic distribution. In contrast, expression of pathogenic variant SMS2$^{I62S}$ or SMS2$^{M64R}$ in each case caused EqtSM$_{cyto}$ to accumulate in numerous puncta that were dispersed throughout the cytosol (*Figure 6A*; *Appendix 1—figure 4A*). Formation of these puncta required expression of a catalytically active pathogenic variant and was not observed when using SM binding-defective cytosolic reporter EqtSol$_{cyto}$ (*Appendix 1—figure 4B*). To check whether translocation of EqtSM to cytosolic puncta is a specific feature of this particular SM-binding toxin, we next used lysenin as alternative cytosolic SM reporter (LysSM$_{cyto}$; *Ellison et al., 2020*). Cells expressing pathogenic SMS2 variants mobilized GFP-tagged LysSM$_{cyto}$ to cytosolic puncta that colocalized extensively with mKate-tagged EqtSM$_{cyto}$ (*Figure 6B, C*;

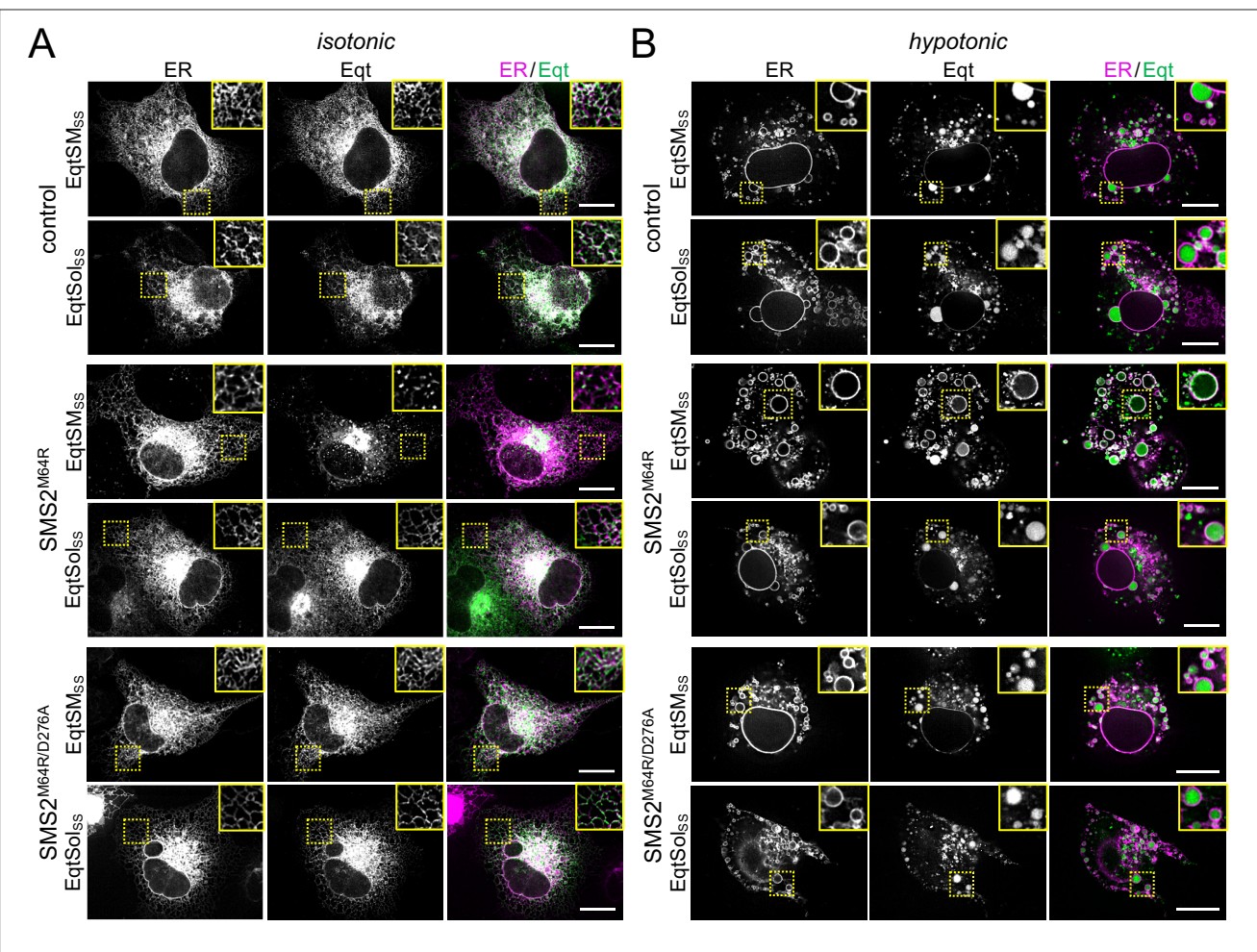

**Figure 5.** Luminal SM reporter EqtSM$_{SS}$ enables visualization of an ER-resident SM pool in SMS2$^{M64R}$-expressing cells. (**A**) Human osteosarcoma U2OS cells co-transfected with mCherry-tagged VAPA (ER, *magenta*) and empty vector (control), SMS2$^{M64R}$ or SMS2$^{M64R/D276A}$ and luminal GFP-tagged SM reporter EqtSM$_{SS}$ or its SM binding-defective derivative, EqtSol$_{SS}$ (Eqt, *green*), were incubated in isotonic medium (100% Optimem) for 5 min and imaged by spinning disc confocal microscopy. (**B**) Cells treated as in (**A**) were incubated in hypotonic medium (1% Optimem) for 5 min and then imaged by spinning disc confocal microscopy. Scale bar, 10 µm.

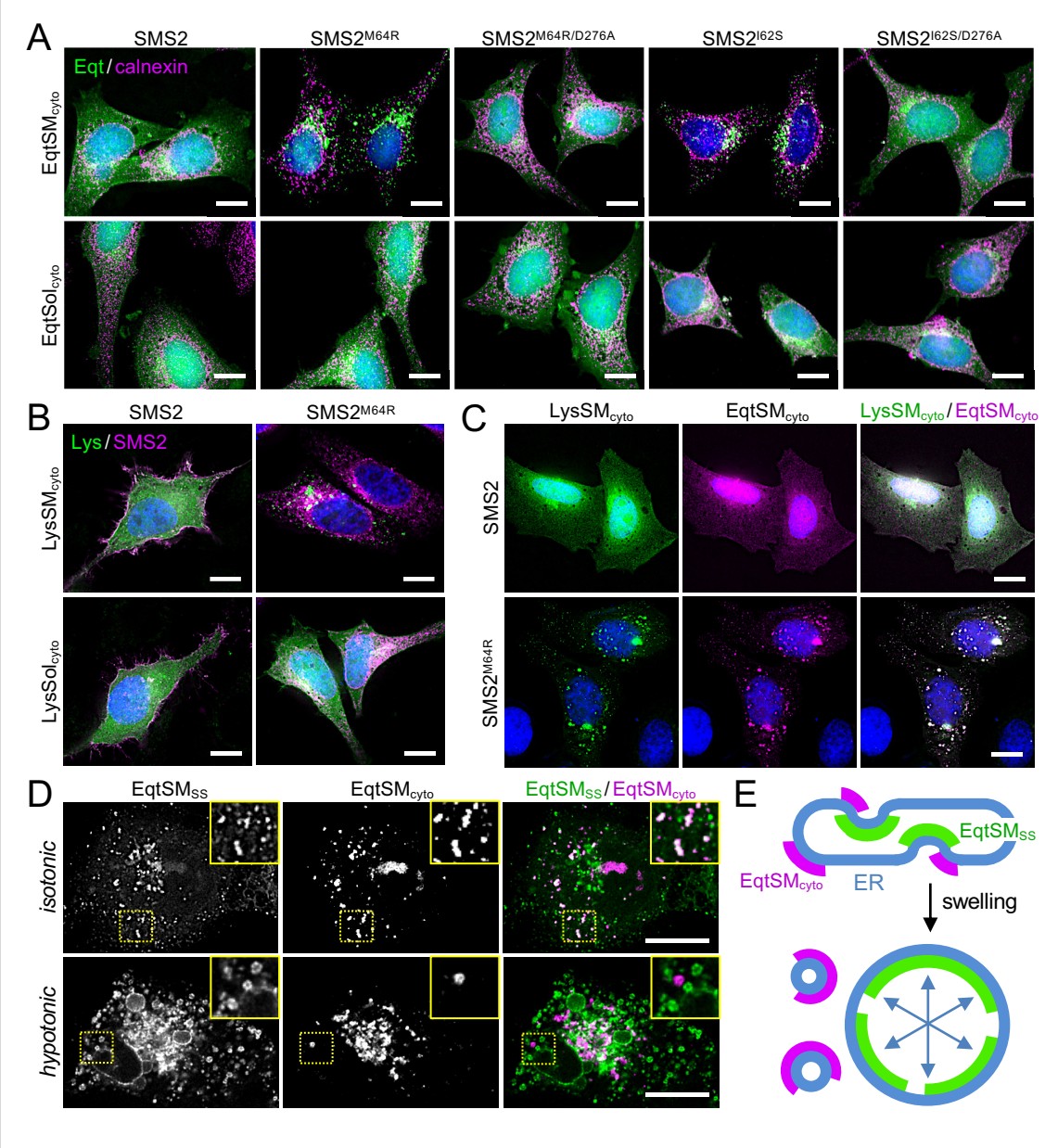

**Figure 6.** Pathogenic SMS2 variants disrupt transbilayer SM asymmetry. (**A**) HeLa ΔSMS1/2 cells transduced with doxycycline-inducible SMS2, SMS2^M64R, SMS2^I62S or their enzyme-dead isoforms (D276A) were transfected with cytosolic GFP-tagged SM reporter EqtSM_cyto or its SM binding-defective derivative, EqtSol_cyto (Eqt, *green*). After treatment with doxycycline (1 μg/ml, 16 h), cells were fixed, immunostained with α-calnexin antibodies (*magenta*), counterstained with DAPI (*blue*) and imaged by DeltaVision microscopy. (**B**) HeLa ΔSMS1/2 cells transduced with doxycycline-inducible SMS2 or SMS2^M64R were transfected with cytosolic GFP-tagged SM reporter LysSM_cyto (*green*) or its SM binding-defective derivative, LysSol_cyto (Lys, *green*). After treatment with doxycycline (1 μg/ml, 16 hr), cells were fixed, immunostained with anti-FLAG antibodies (SMS2, *magenta*), counterstained with DAPI (*blue*) and imaged by DeltaVision microscopy. Scale bar, 10 μm. (**C**) HeLa ΔSMS1/2 cells transduced with doxycycline-inducible SMS2 or SMS2^M64R were co-transfected with GFP-tagged LysSM_cyto (*green*) and mKate-tagged EqtSM_cyto (*magenta*). After treatment with doxycycline (1 μg/ml, 16 hr), cells were fixed, counterstained with DAPI (*blue*) and imaged by DeltaVision microscopy. Scale bar, 10 μm. (**D**) U2OS cells co-transfected with SMS2^M64R, luminal GFP-tagged EqtSM_SS (*green*) and cytosolic mKate-tagged EqtSM_cyto (*magenta*) were incubated in isotonic (100% Optimen) or hypotonic media (1% Optimem) for 5 min, and then imaged by spinning disc confocal microscopy. Scale bar, 10 μm. (**E**) Graphic illustration of the data shown in (**D**). See text for details.

*Appendix 1—figure 4C*). Such puncta did not form in cells expressing wildtype SMS2 or when using a non-SM binding lysenin variant (LysSol$_{cyto}$). These results indicate that pathogenic SMS2 variants render SM accessible to cytosolic SM reporters, presumably because ER-resident scramblases enable SM produced by these variants to readily equilibrate across the ER bilayer.

Remarkably, the EqtSM$_{cyto}$-positive puncta in SMS2$^{I62S}$ or SMS2$^{M64R}$-expressing cells largely segregated from a wide range or organellar markers, including VAPA (ER), Sec16L (ER exit sites), ERGIC-53 (*cis*-Golgi), GM130 (*medial*-Golgi), TGN64 (*trans*-Golgi), EEA1 (early endosomes), LAMP1 (lysosomes), and LD540 (lipid droplets; *Appendix 1—figure 5*). However, EqtSM$_{cyto}$-positive puncta displayed considerable overlap with EqtSM$_{SS}$-positive puncta formed in the ER lumen of SMS2$^{M64R}$-expressing cells (*Figure 6D*, upper row), suggesting that the former originate at least in part from the ER. Upon hypotonic swelling, the EqtSM$_{SS}$-positive puncta became largely continuous with the ER membrane and segregated from the EqtSM$_{cyto}$-positive puncta (*Figure 6D*, bottom row). Under these conditions, EqtSM$_{cyto}$-positive puncta gave rise to small vesicular structures, implying the presence of a membranous core. Based on these results, we envision that SM synthesized by pathogenic SMS2 variants readily equilibrates across the ER bilayer, triggering mobilization of both luminal and cytosolic EqtSM (*Figure 6E*). Whether condensation of EqtSM into puncta reflects binding of the reporter to pre-existing SM microdomains or involves a clustering of SM-bound reporters driven by a residual pore-forming activity remains to be established. Conceivably, binding of EqtSM to the membrane may alter its curvature and even pinch off SM-enriched membrane regions, analogous to the shedding of extracellular vesicles observed when cells are exposed to the cholesterol/SM binding protein ostreolysin A (*Skočaj et al., 2016*). In fact, such shedding may explain why EqtSM$_{cyto}$-positive puncta do not co-localize with a wide range of organellar markers (*Appendix 1—figure 5*). While our findings underline the value of EqtSM and LysSM as tools for probing SM topology in organellar bilayers, further studies are warranted to address their potential impact on the structural organization of SM-rich membranes.

Based on the foregoing, we reasoned that cells expressing pathogenic SMS2 variants may have a diminished capacity to concentrate SM on their surface. To challenge this idea, we stained the surface of intact wildtype or ΔSMS1/2 cells expressing SMS2 or SMS2$^{M64R}$ with recombinant EqtSM. Cell surface labeling was visualized by fluorescence microscopy and quantitatively assessed by flow cytometry. As shown in *Figure 7A*, wildtype cells could be readily stained with the SM reporter whereas ΔSMS1/2 cells were devoid of EqtSM staining. Expression of SMS2, but not enzyme-dead SMS2$^{D276A}$, restored EqtSM staining of ΔSMS1/2 cells to a level approaching that of wildtype cells. In contrast, expression of SMS2$^{M64R}$ restored cell surface staining to only a minor degree (*Figure 7A, B*) even though the PM-associated SM pool of these cells was close to that of wildtype or SMS2-expressing ΔSMS1/2 cells (*Figure 4D*; *Figure 4—source data 2*). Taken together, our findings suggest that pathogenic SMS2 variants disrupt the establishment of transbilayer SM asymmetry.

## Pathogenic SMS2 variants affect membrane lipid order along the secretory pathway

Owing to its saturated nature and affinity for sterols, SM contributes significantly to lipid order in cellular membranes. As pathogenic SMS2 variants enhance SM levels in the ER and undermine the ability of cells to concentrate SM on their surface, we next measured the lipid order at these locations in cells expressing SMS2 or SMS2$^{M64R}$ using two Nile Red (NR)-based solvatochromic probes, NR12A and NRER$_{Cl}$ (*Danylchuk et al., 2021*; *Danylchuk et al., 2019*). The emission spectra of these probes are blue-shifted in tightly packed lipid bilayers, which is due to a reduced polarity in the nano-environment of the NR fluorophore. The relative extent of these spectral shifts can be quantified in cellular membranes using ratiometric imaging. In NR12A, the presence of a charged membrane anchor group that blocks passive flip-flop across membrane bilayers makes this probe ideally suited to selectively quantify lipid order in the outer PM leaflet of live cells when added to the extracellular medium (*Danylchuk et al., 2019*). On the other hand, a propyl chloride group in NRER$_{Cl}$ targets this probe to the ER (*Danylchuk et al., 2021*) even though some residual staining of other organelles cannot be excluded. As expected, the lipid order reported by NR12A in the outer PM leaflet of SM-deficient ΔSMS1/2 cells was drastically reduced in comparison to that of wildtype cells (*Figure 7C, D - Figure 7—source data 1*). Expression of SMS2 partially restored lipid order. In contrast, expression of SMS2$^{M64R}$ failed to restore lipid order to any appreciable degree. Conversely, the lipid order reported by NRER$_{Cl}$ in SMS2$^{M64R}$-expressing cells was significantly enhanced in comparison to that of

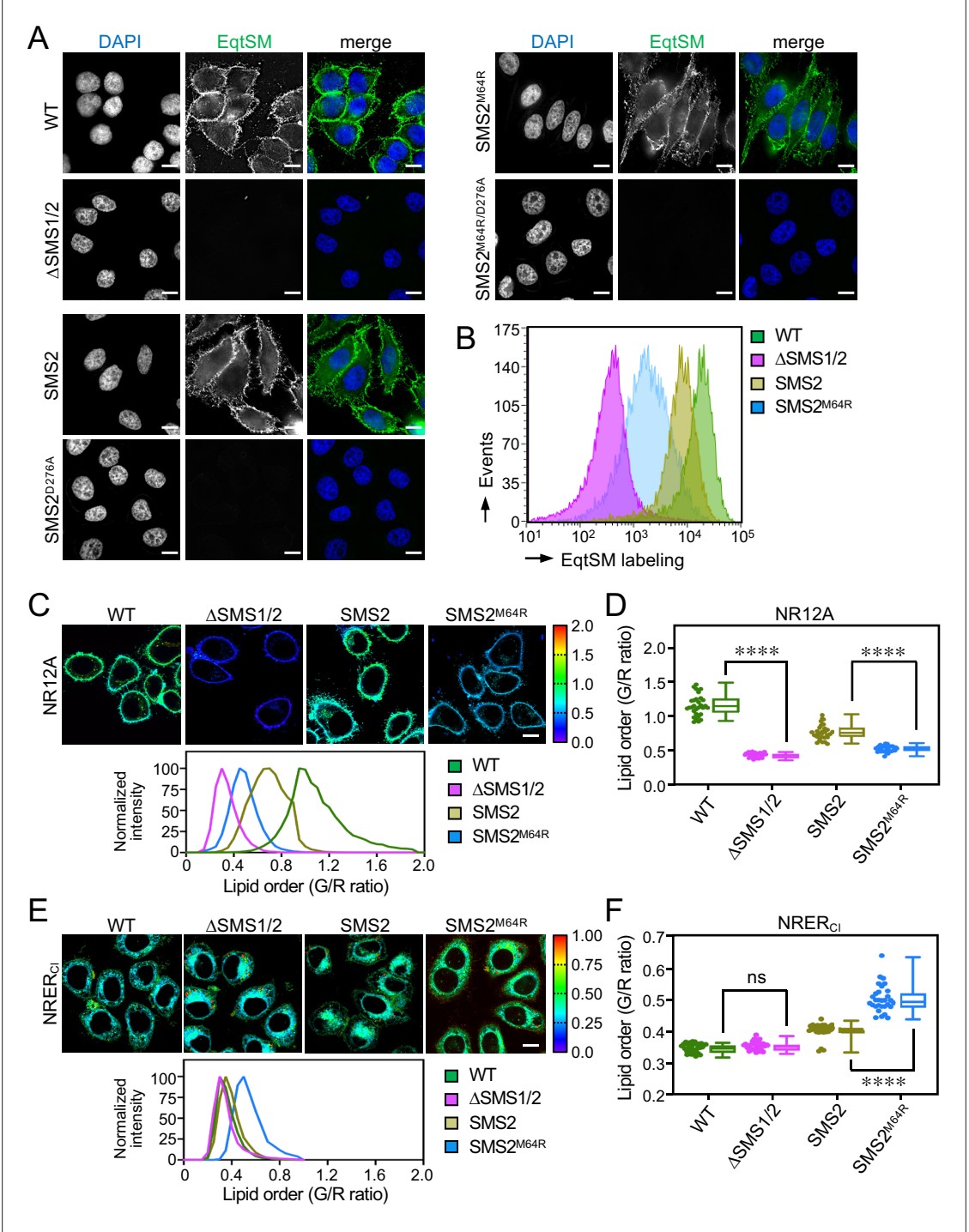

**Figure 7.** SMS2[M64R]-expressing cells fail to concentrate SM on their surface and exhibit imbalances in lipid order. (**A**) HeLa wildtype (WT) or ΔSMS1/2 cells transduced with doxycycline-inducible SMS2, SMS2[M64R] or their enzyme-dead isoforms (D276A) were treated with doxycycline (1 μg/ml, 16 h), incubated with FLAG-tagged EqtSM, fixed, co-stained with α-FLAG antibody (*green*) and DAPI (*blue*), and imaged by DeltaVision microscopy. (**B**) Cells treated as in (**A**) were analyzed by flow cytometry to quantitatively assess EqtSM labeling of their surface. (**C**) Cells treated as in (**A**) were stained with 0.2 μM NR12A for 10 min and analyzed by ratiometric fluorescence microscopy to probe the lipid order in the outer PM leaflet. Warmer colors reflect a higher lipid order. (**D**) Quantitative assessment of changes in lipid order in the outer PM leaflet of cells treated as in (**C**). n=30 cells per condition over two biologically independent experiments. (**E**) Cells treated as in (**A**) were stained with 0.2 μM NRER[Cl] for 10 min and analyzed by ratiometric fluorescence microscopy to probe lipid order in the ER. Warmer colors reflect a higher lipid order. (**F**) Quantitative assessment of changes in lipid order

*Figure 7 continued on next page*

*Figure 7 continued*

in the ER of cells treated as in (**E**). n=30 cells per condition over two biologically independent experiments. All p values calculated by unpaired *t*-test. Scale bar, 10 μm.

The online version of this article includes the following source data for figure 7:

**Source data 1.** Raw data of the quantitative analysis of ratiometric fluorescence microscopy images captured under the conditions indicated.

SMS2-expressing cells (*Figure 7E, F - Figure 7—source data 1*). Hence, the perturbation of subcellular SM distributions caused by pathogenic SMS2 variants is accompanied by major imbalances in lipid order along the secretory pathway.

## Pathogenic SMS2 variants perturb subcellular cholesterol pools

As preferred cholesterol interaction partner, SM directly participates in the subcellular organization of cholesterol (*Das et al., 2014*; *Slotte, 2013*). Therefore, it was surprising that the marked accumulation of SM in the ER of SMS$^{M64R}$-expressing cells had no obvious impact on ER-bound cholesterol levels or the cellular pool of cholesteryl esters (CE; *Figure 3D*; *Figure 3—source data 2*). To examine whether pathogenic SMS2 variants influence cholesterol organization in cells, we used a mCherry-tagged D4H sterol reporter derived from the perfringolysin O $\theta$-toxin of *Clostridium perfringens*. This reporter recognizes cholesterol when present at >20 mol% in membranes and mainly decorates the inner leaflet of the PM at steady state when expressed as cytosolic protein. Part of the reason for the high detection threshold is that the reporter detects free cholesterol in the membrane but not cholesterol in complex with SM (*Das et al., 2014*). Moreover, a previous study showed that plasmalemmal PS is essential for retaining D4H-accessible cholesterol in the inner PM leaflet (*Maekawa and Fairn, 2015*). Accordingly, we found that cytosolic D4H-mCherry primarily stained the inner PM leaflet in wildtype HeLa cells. In contrast, the probe displayed a more diffuse cytosolic distribution in ΔSMS1/2 cells (*Appendix 1—figure 6A*). Expression of SMS2 inΔSMS1/2 cells restored PM staining, indicating that PM-associated SM is critical for controlling the reporter-accessible cholesterol pool in the inner PM leaflet (*Appendix 1—figure 6B*). Strikingly, in ΔSMS1/2 cells expressing pathogenic variant SMS2$^{M64R}$ or SMS2$^{I62S}$, cytosolic D4H-mCherry did not label the inner PM leaflet but primarily accumulated on intracellular vesicles that were largely segregated from EqtSM$_{cyto}$-positive puncta (*Figure 8A*; *Appendix 1—figure 6B*). These vesicles co-localized extensively with dextran-positive endolysosomal compartments (*Figure 8B*). Whether redistribution of cytosolic D4H-mCherry from the PM to endolysosomes induced by pathogenic SMS2 variants is due to an enhanced Golgi-to-endosome trafficking of cholesterol or an impaired cholesterol export from lysosomes remains to be established. As PM cholesterol levels are largely unaffected (*Figure 4D*; *Figure 4—source data 2*), it is conceivable that the presence of SM in the inner PM leaflet of cells expressing pathogenic SMS2 variants may render a coexisting cholesterol pool inaccessible to the reporter.

Since cholesterol has a stronger affinity for SM than for PS or other phospholipid classes, altering the SM concentration affects the behavior of cholesterol in artificial and biological membranes (*Slotte, 2013*). For instance, when exposed to the cholesterol-absorbing agent methyl-β-cyclodextrin (mβCD), SM-depleted cells readily lose PM-associated cholesterol and consequently lose their viability more rapidly than wildtype cells (*Fukasawa et al., 2000*; *Hanada et al., 2003*). As complementary approach to determine the impact of pathogenic SMS2 variants on cholesterol organization in the PM, we next probed ΔSMS1/2 cells expressing wildtype or pathogenic SMS2 variants for their sensitivity toward mβCD. As expected, ΔSMS1/2 cells displayed a substantially reduced tolerance for mβCD in comparison to wildtype cells (*Figure 8C*; *Figure 8—source data 1*). Expression of SMS2 restored mβCD tolerance of ΔSMS1/2 cells to that of wildtype cells. In contrast, expression of SMS$^{M64R}$ or SMS2$^{I62S}$ in each case failed to render ΔSMS1/2 cells resistant toward mβCD. These results provide additional support for the notion that pathogenic SMS2 variants significantly affect cholesterol organization in the PM.

## Patient-derived fibroblasts display imbalances in SM distribution and lipid packing

We next asked whether the aberrant SM and cholesterol distributions observed upon heterologous expression of pathogenic SMS2 variants also occur in cells of patients with OP-CDL. To address this, skin fibroblasts derived from patients with the missense variant p.I62S or p.M64R and healthy controls

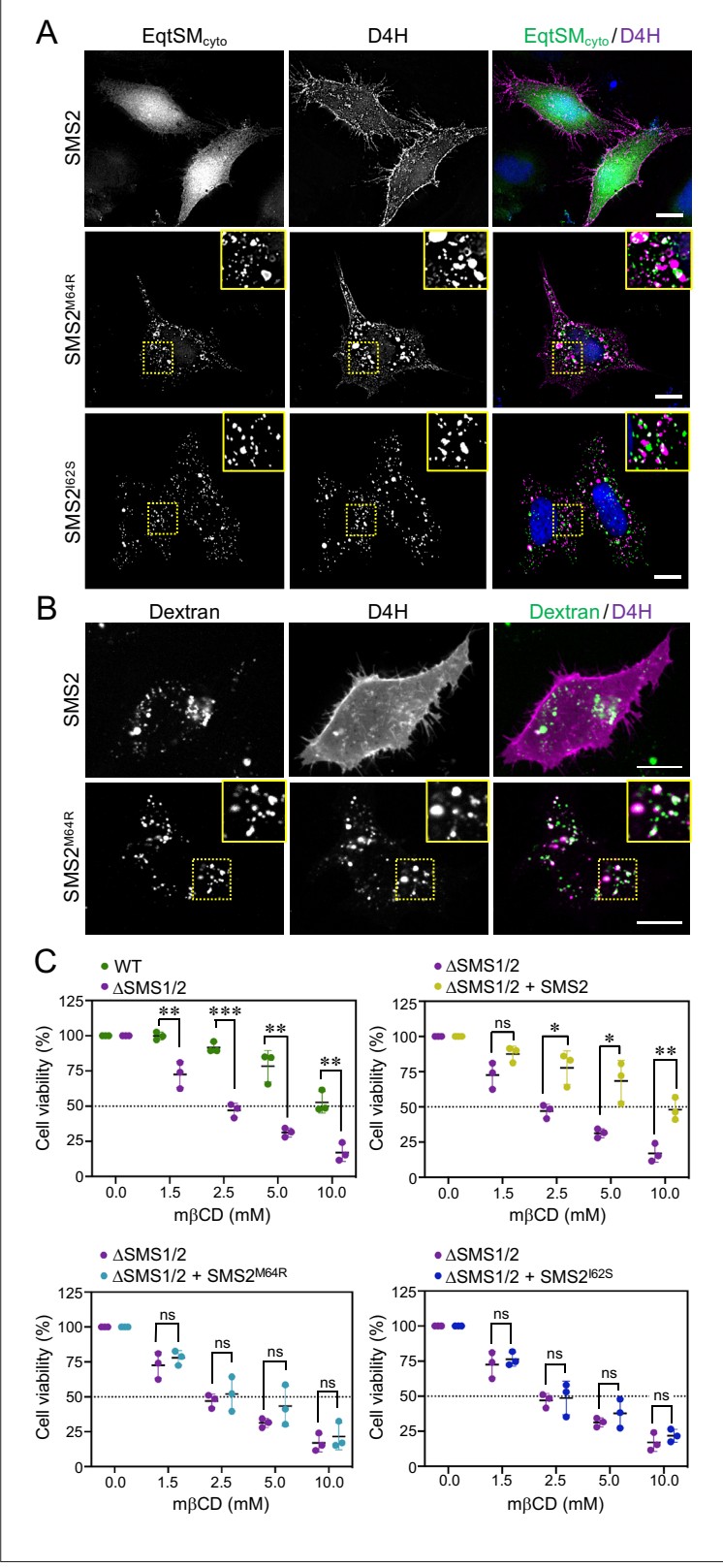

**Figure 8.** Pathogenic SMS2 variants perturb subcellular cholesterol pools. (**A**) HeLa ΔSMS1/2 cells transduced with doxycycline-inducible SMS2, SMS2^{M64R} or SMS2^{I62S} were co-transfected with GFP-tagged EqtSM_{cyto} (*green*) and mCherry-tagged cytosolic sterol reporter D4H (*red*). Next, cells were treated with 1 μg/ml doxycycline for 16 hr, fixed, counterstained with DAPI (*blue*) and visualized by DeltaVision microscopy. (**B**) HeLa ΔSMS1/2 cells

*Figure 8 continued on next page*

*Figure 8 continued*

stably transduced with FLAG-tagged SMS2 or SMS2$^{M64R}$ were transfected with mCherry-tagged D4H (*red*), labeled with fluorescein-conjugated dextran (*green*) in the presence of 1 µg/ml doxycycline for 16 hr and then imaged by spinning disc confocal microscopy. (**C**) HeLa wildtype (WT) or ΔSMS1/2 cells stably transduced with SMS2, SMS2$^{M64R}$ or SMS2$^{I62S}$ were treated with 1 µg/ml doxycycline for 16 hr. Next, cells were exposed to the indicated concentration of methyl β-cyclodextrin (mβCD) for 1 hr and cell viability was assessed using Prestoblue reagent. Data shown are averages of four technical replicates from n=3 biological replicates. *p<0.05, **p<0.01, ***p<0.001 by paired *t* test. Scale bar, 10 µm.

The online version of this article includes the following source data for figure 8:

**Source data 1.** Raw data of the quantitative analysis of cell viability under the conditions indicated.

---

were co-transfected with the luminal SM reporter EqtSM$_{SS}$ and mCherry-tagged VAPA as ER marker. Next, the fibroblasts were subjected to hypotonic swelling and live cell imaging. Strikingly, in patient fibroblasts the membranes of ER-derived vesicles were extensively labelled with EqtSM$_{SS}$ (***Figure 9A***). In contrast, in fibroblasts of healthy controls, EqtSM$_{SS}$ was found exclusively in the lumen of ER-derived vesicles. This indicates that the ER in patient fibroblasts contains substantially elevated SM levels. In addition, we found that the cytosolic SM reporter EqtSM$_{cyto}$ accumulated in numerous puncta when expressed in patient fibroblasts while its expression in fibroblasts of healthy controls resulted in a diffuse cytosolic distribution (***Figure 9B***). Formation of Eqt-positive puncta in patient fibroblasts did not occur when using the SM binding-defective reporter, EqtSol$_{cyto}$. Thus, besides accumulating SM in the ER, patient fibroblasts display a breakdown of transbilayer SM asymmetry. Interestingly, these aberrant SM distributions were accompanied by significant alterations in lipid order on the cell surface (***Figure 9C***) and in the ER (***Figure 9D***; ***Figure 9—source data 1***). Moreover, while the cytosolic cholesterol reporter D4H-mCherry primarily stained the PM of fibroblasts of healthy controls, in patient fibroblasts a substantial portion of the reporter was shifted to intracellular vesicles (***Appendix 1— figure 7***). This indicates that pathogenic SMS2 variants p.I62S and p.M64R, which underly a spectrum of severe skeletal conditions, affect the subcellular organization of SM and cholesterol to an extend large enough to impact on the lipid order along membranes of the secretory pathway.

## Discussion

SM in mammalian cells is specifically enriched in the exoplasmic leaflets of the PM, the *trans*-Golgi and endolysosomal organelles. While maintenance of its nonrandom subcellular distribution is thought to be relevant for a variety of physiological processes, experimental proof for this concept is scarce. Here, we show that inborn pathogenic SMS2 variants p.M64R and p.I62S identified in patients with a severe form of OP-CDL cause profound perturbations in subcellular SM organization. Both variants retain full enzymatic activity but are unable to leave the ER owing to a defective autonomous ER export signal in their *N*-terminal cytosolic tail. Consequently, bulk SM production is mistargeted to the ER, the site for de novo synthesis of the SM precursor ceramide. Cells expressing pathogenic SMS2 variants accumulate PM-like SM levels in the ER and display an aberrant mobilization of cytosolic SM reporters, signifying a loss of transbilayer SM asymmetry presumably due to a constitutive SM scrambling across the ER bilayer (***Figure 9E***). From this we infer that pathogenic SMS2 variants in fact abolish two types of SM gradients: one running along the secretory pathway and the other one across the bilayers of secretory organelles. These marked deviations in SM distribution also occur in OP-CDL patient fibroblasts and are accompanied by significant imbalances in cholesterol organization, glycerophospholipid profiles and membrane lipid order in the secretory pathway. Based on these findings, we postulate that pathogenic SMS2 variants undermine the capacity of osteogenic cells to uphold nonrandom lipid distributions that are critical for their bone forming activity.

As SM is the preferred interaction partner of cholesterol (***Slotte, 2013***), bulk production of SM in the *trans*-Golgi and its sorting into anterograde transport vesicles would promote formation of a cholesterol gradient along the secretory pathway. However, we now find that cells harboring pathogenic SMS2 variants retain the ability to concentrate cholesterol in the PM and keep their ER levels low in spite of a dissipated SM gradient. Even though cultured cells may not make much cholesterol in the ER and instead obtain bulk cholesterol from lipoproteins in the culture medium, previous work revealed that a stream of cholesterol constantly travels from the PM to the ER (***Infante and***

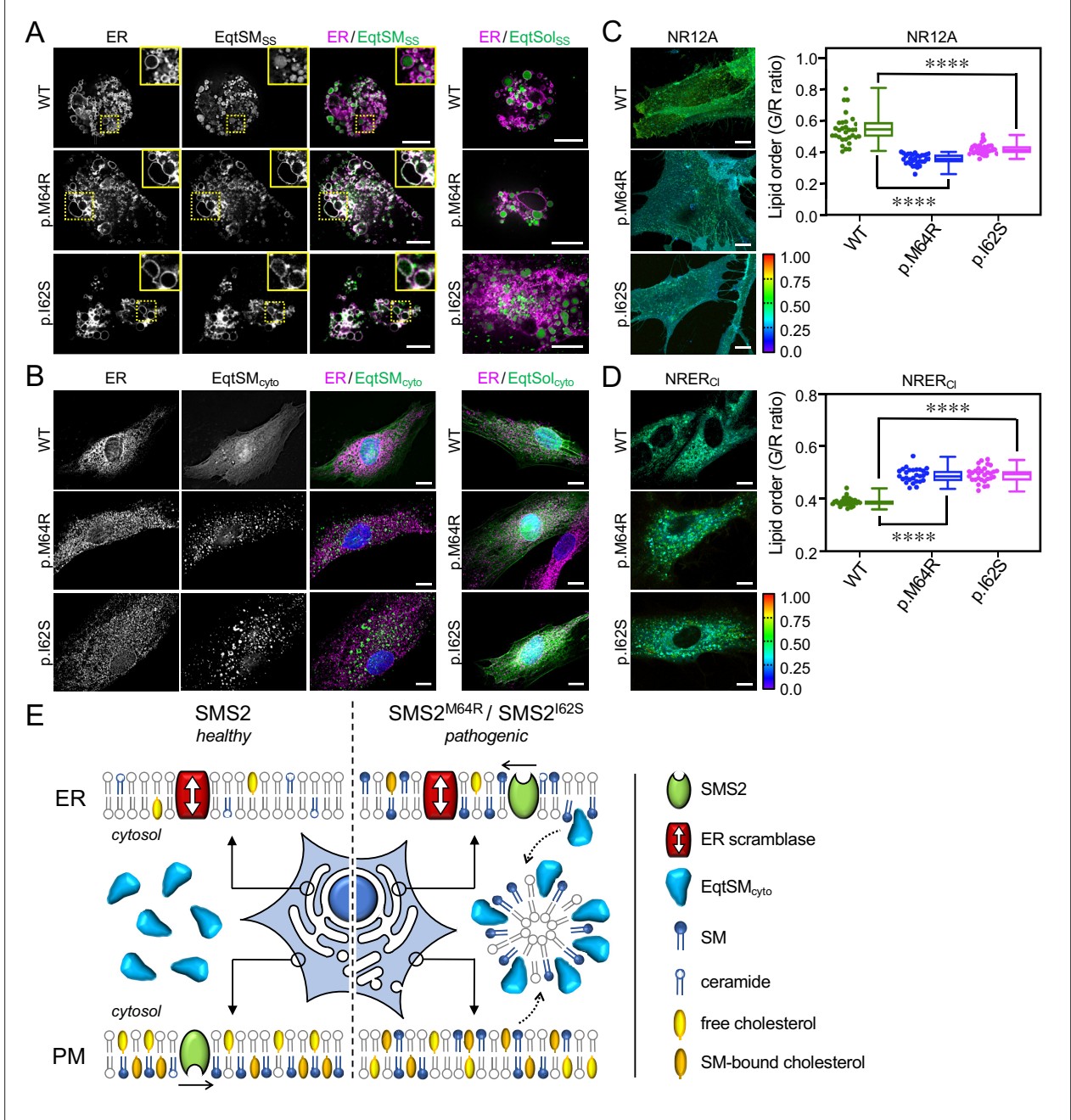

**Figure 9.** Patient-derived fibroblasts display perturbations in SM distribution and lipid order. (**A**) Control (WT) or patient-derived human skin fibroblasts carrying heterozygous missense variants c.185T>G (p.I62S) or c.191T>G (p.M64R) in *SGMS2* were co-transfected with mCherry-tagged VAPA (ER, *magenta*) and GFP-tagged EqtSM$_{SS}$ (*green*). Co-transfections with GFP-tagged EqtSol$_{SS}$ served as control. After 16 hr, cells were incubated in hypotonic medium (1% Optimem) for 5 min and imaged by spinning disc confocal microscopy. (**B**) Fibroblasts as in (**A**) were transfected with GFP-tagged EqtSM$_{cyto}$. After 16 hr, cells were fixed, immunostained with α-calnexin antibodies (ER, *magenta*), counterstained with DAPI (*blue*) and imaged by DeltaVision microscopy. (**C**) Fibroblasts as in (**A**) were stained with 0.2 μM NR12A for 10 min and analyzed by ratiometric fluorescence microscopy to quantitatively access lipid order in the outer PM leaflet. Warmer colors reflect a higher lipid order. n=30 cells per condition analyzed over two biologically independent experiments. (**D**) Fibroblasts as in (**A**) were stained with 0.2 μM NRER$_{Cl}$ for 10 min and analyzed by ratiometric fluorescence microscopy to quantitatively access lipid order in the ER. Warmer colors reflect a higher lipid order. n=27 cells per condition analyzed over two biologically independent experiments. All p values calculated by unpaired *t*-test. Scale bar, 10 μm. (**E**) Graphic illustration of how pathogenic SMS2 variants affect the subcellular SM distribution, promote translocation of EqtSM$_{cyto}$ to cytosolic puncta, and influence cholesterol organization at the PM. See text for details.

The online version of this article includes the following source data for figure 9:

**Source data 1.** Raw data of the quantitative analysis of ratiometric fluorescence microscopy images captured under the conditions indicated.

*Radhakrishnan, 2017*). This implies that cells are equipped with an effective mechanism to prevent a toxic rise of cholesterol in ER bilayers with an abnormally high SM content. One mechanism for removing excess cholesterol from the ER involves its esterification and storage in lipid droplets. However, pathogenic SMS2 variants had no impact on the cellular pool of cholesteryl esters. An alternative mechanism involves the oxysterol binding protein OSBP, which mediates net transfer of cholesterol from the ER to the *trans*-Golgi by counter transporting phosphatidylinositol-4-phosphate (PI4P), a lipid continuously produced in the *trans*-Golgi and turned over in the ER (*Mesmin et al., 2013*). As critical determinant of intracellular cholesterol flows, the cholesterol/PI4P exchange activity of OSBP also influences membrane lipid order in the secretory pathway (*Mesmin et al., 2017*). Hence, future studies addressing whether pathogenic SMS2 variants cause an upregulation of the PI4P-consuming OSBP cycle to counteract thermodynamic trapping of cholesterol by an expanding SM pool in the ER may prove fruitful. The PI4P-dependent countertransport mechanism found to energize net transfer of cholesterol also helps drive PS export from the ER to create a PS gradient along the secretory pathway (*Chung et al., 2015*; *Moser von Filseck et al., 2015*). In contrast, the PM-like SM content of the ER in cells harboring pathogenic SMS2 variants indicates that such mechanism does not exist for SM and that the ER lacks transport systems to efficiently distribute this lipid to other organelles in sync with its rate of synthesis.

Membrane biogenesis in the ER requires cross-bilayer movement of phospholipids, which is mediated by ER-resident scramblases (*Ghanbarpour et al., 2021*; *Huang et al., 2021*; *Pomorski and Menon, 2016*). These scramblases display low specificity, with phospholipids and sphingolipids being translocated with similar kinetics (*Buton et al., 2002*; *Chalat et al., 2012*). Consequently, SM produced by pathogenic SMS2 variants in the luminal leaflet of the ER should readily equilibrate with the cytosolic leaflet. Indeed, both p.M64R and p.I62S variants triggered mobilization of a cytosolic SM reporter. Moreover, cells expressing SMS2$^{M64R}$ had similar levels of SM in the PM as controls but showed a significantly reduced SM reporter staining of their surface, indicative of a perturbed SM asymmetry across the PM. Consistent with a reduced SM pool in the exoplasmic leaflet, these cells displayed a lower lipid packing on their surface and lost their viability more rapidly when exposed to a cholesterol-absorbing agent than controls. Consistent with an elevated SM pool in the cytosolic leaflet, cells harboring pathogenic SMS2 variants displayed a reduced PM staining with a cytosolic sterol reporter that recognizes free cholesterol but not cholesterol sequestered by SM. Thus, while pathogenic SMS2 variants have no impact on PM cholesterol levels, our data suggest that they affect the equilibrium between active and SM-bound cholesterol pools on both sides of the PM (*Figure 9E*).

Our study also yields insights into how cells cope with a major deviation in the lipid composition of the ER. The dramatic rise in ER-associated SM levels caused by pathogenic SMS2 variants was accompanied by a marked increase in PC desaturation and a nearly twofold expansion of the PE pool. While it remains to be established how these alterations are implemented, it is conceivable that an enhanced desaturation and rise in cone-shaped, ethanolamine-containing phospholipids are part of an adaptive cellular response to counter an SM-mediated rigidification of the ER bilayer and preserve the organelle's central role in membrane biogenesis and secretion. Strikingly, pathogenic SMS2 variants also caused a sharp rise in ER-associated Cer1P levels. Cer1P is produced by ceramide kinase CERK and functions as a key signaling lipid in the regulation of cell growth, survival and inflammation (*Presa et al., 2020*). In addition to stimulating production of pro-inflammatory cytokines through direct activation of a specific cytosolic phospholipase (*Lamour et al., 2009*), Cer1P promotes cell survival at least in part by blocking enzymes involved in ceramide production (*Granado et al., 2009*). Our present findings indicate that Cer1P production is tightly coupled to SM biosynthesis. The prospect that CERK-mediated Cer1P formation serves a role in the mechanism by which cells sense and respond to imbalances in the lipid composition of their secretory organelles merits further consideration.

While this study focuses on the impact of the missense SMS2 variants p.M64R and p.I62S on the subcellular organization of SM, the nonsense pathogenic SMS2 variant p.Arg50* is more common in patients with OP-CDL and associated with a milder bone phenotype (*Mäkitie et al., 2021*). We originally anticipated that this variant encodes a severely truncated and enzyme-dead version of SMS2 that contains only part of the *N*-terminal cytosolic tail (*Pekkinen et al., 2019*). However, our recent work revealed that Met64 in the p.Arg50* variant is utilized as alternative start codon, yielding a truncated but catalytically active enzyme that is mislocalized to the *cis/medial* Golgi (T. Sokoya and J. Holthuis,

unpublished data). Unlike pathogenic variants SMS2$^{I62S}$ and SMS2$^{M64R}$, SMS2$^{R50X}$ would have no direct access to ER-derived ceramides and must compete with GlcCer synthase for ceramides delivered to the *cis/medial* Golgi. Backflow of SM produced in the *cis/medial* Golgi to the ER may affect the lipid landscape in the secretory pathway of cells expressing SMS2$^{R50X}$, but to a lesser extent than that observed in cells expressing the pathogenic variants SMS2$^{I62S}$ and SMS2$^{M64R}$.

Our findings raise the question how a disrupted subcellular organization of SM leads to osteo-porosis and skeletal dysplasia. A quantitative analysis of SMS2 transcript levels in a murine tissue panel revealed the highest expression in cortical bone and vertebrae (*Pekkinen et al., 2019*). This implies that the impact of pathogenic SMS2 variants on the lipid composition of secretory organelles will be most severe in bone cells of the affected individuals. Bone formation involves deposition of collagen fibrils into a matrix and its subsequent mineralization. Interestingly, pathogenic variants of core components of COPII-coated vesicles have been reported to cause craniofacial and skeletal defects by selectively disrupting procollagen export from the ER (*Boyadjiev et al., 2006*; *Garbes et al., 2015*). Moreover, loss of TANGO1, an ER-resident transmembrane protein required for pack-aging the bulky procollagen fibers into COPII vesicles, results in neonatal lethality due to insufficient bone mineralization (*Guillemyn et al., 2021*). It is conceivable that ER export of procollagen is partic-ularly susceptible to the bilayer rigidifying effect of SM synthesized by pathogenic SMS2 variants.

However, an alternative scenario is that a perturbed PM SM asymmetry in osteogenic cells of OP-CDL patients negatively affects bone mineralization. This process involves matrix vesicles that bud off from the apical membrane of osteoblasts and deposit their Ca$^{2+}$ and phosphate-rich content where matrix mineralization is propagated (*Murshed, 2018*). Bone mineralization critically relies on neutral SMase-2 (SMPD3), another membrane-bound enzyme highly expressed in bone that cleaves SM in the cytosolic leaflet of the PM to generate ceramide and phosphocholine (*Aubin et al., 2005*). nSMase-2 gains access to SM upon SM scrambling by the Ca$^{2+}$-activated scramblase TMEM16F (*Niekamp et al., 2022*). Loss of TMEM16F leads to decreased mineral deposition in skeletal tissues (*Ehlen et al., 2013*), suggesting that this process may require a TMEM16F-mediated supply of exoplasmic SM to nSMase-2 for SM hydrolysis in the cytosolic leaflet. This arrangement may serve to ensure a contin-uous supply of phosphocholine as a source of phosphate required for normal bone mineralization (*Pekkinen et al., 2019*). By disrupting SM asymmetry, pathogenic SMS2 variants may interfere with a coordinated release of the lipid-based phosphate store, thus impairing normal bone mineralization.

In sum, the present study indicates that bone critical SMS2 variants p.M64R and p.I62S exert their pathogenic effects by redirecting bulk SM production to the ER, thereby causing a wide-ranging perturbation of lipid distributions and membrane properties along the secretory pathway. Besides highlighting how cells respond to a major assault on the lipid code of the ER, our findings provide important insights into the pathobiochemistry underlying OP-CDL.

## Materials and methods
### Chemical reagents
Chemical reagents used were: doxycycline (Sigma Aldrich, D891), puromycin (Sigma Aldrich, P8833), polybrene (Sigma-Aldrich, TR-1003), methyl-β-cyclodextrin (Sigma Aldrich, C4555), G418 (Sigma-Aldrich, G8168), and 3-azido-7-hydroxycoumarin (Jena Bioscience, CLK-FA047). LD540 dye was a kind gift from Christoph Thiele (University of Bonn, Germany) and described in *Spandl et al., 2009*. NR12A and NR-ER$_{cl}$ were synthesized as described in *Danylchuk et al., 2021*.

### Antibodies
Antibodies used were: rabbit polyclonal anti-calnexin (Santa Cruz, sc-11397; IB 1:1000), goat poly-clonal anti-calnexin (Santa Cruz, sc-6495; IF 1:200, IB 1:1000), rabbit polyclonal anti-β-calnexin (Abcam, ab10286; IP 1:1000), mouse monoclonal anti-FLAG-tag (Abcam, ab205606; IB 1:1000; IF 1:400), mouse monoclonal anti-SMS2 (Santa Cruz, sc-293384; IB 1:1000), mouse monoclonal anti-β-actin (Sigma, A1978; IB 1:50,000), mouse monoclonal anti-mitochondrial surface p60 (Millipore, MAB1273; IB 1:1000), mouse monoclonal anti-Na/K ATPase (Santa Cruz, sc-48345; IB 1:1000), rabbit monoclonal anti-Na/K ATPase (Abcam, ab-76020; IB 1:1,000; IF 1:400), mouse monoclonal anti-ERGIC53 (Novus, np62-03381; IF 1:400), mouse monoclonal anti-GM130 (BD biosciences, 610823; IF 1:400), sheep polyclonal anti-TGN46 (Bio-Rad, AHP1586; IF 1:400), mouse monoclonal anti-EEA1 (Cell Signaling,

48453; IF 1:400), mouse monoclonal anti-LAMP-1 (Santa Cruz, sc-20011; IB 1:1,000; IF 1:400), HRP-conjugated goat anti-rabbit IgG (Thermo Fisher Scientific, 31460; IB 1:5000), HRP-conjugated goat anti-mouse IgG (Thermo Fisher Scientific, 31430; IB 1:5000), HRP-conjugated donkey anti-goat IgG (Thermo Fisher Scientific; PA1-28664; IB 1:5000), Dylight 488-conjugated donkey-anti-sheep/goat IgG (Bio-Rad, STAR88D488GA; IF 1:400), Cyanine Cy2-conjugated donkey anti-mouse IgG (Jackson ImmunoResearch Laboratories, 715-225-150; IF 1:400); Cyanine Cy2-conjugated donkey anti-rabbit IgG (Jackson ImmunoResearch Laboratories, 711-225-152; IF 1:400); Cyanine Cy3-conjugated donkey anti-rabbit IgG (Jackson ImmunoResearch Laboratories, 715-165-152; IF 1:400); Cyanine Cy3-conjugated donkey anti-mouse IgG (Jackson ImmunoResearch Laboratories, 715-165-150; IF 1:400); Cyanine Cy3-conjugated donkey anti-goat IgG (Jackson ImmunoResearch Laboratories, 705-165-147; IF 1:400), Cyanine Cy5-conjugated donkey anti-rabbit IgG (Jackson ImmunoResearch Laboratories, 711-175-152; IF 1:400) and Cyanine Cy5-conjugated donkey anti-goat IgG (Jackson ImmunoResearch Laboratories, 705-175-147; IF 1:400).

## DNA constructs

pcDNA3.1(+) encoding *N*-terminal FLAG-tagged SMS2, SMS2$^{I62S}$ and SMS2$^{M64R}$ were described in *Pekkinen et al., 2019*. DNA encoding *N*-terminal FLAG-tagged chimera SMSr-SMS2$_{11-77}$ was synthetically prepared (IDT, Belgium) and inserted into pcDNA3.1(+) using BamHI and NotI restriction sites. Pathogenic mutations were introduced using a QuickChangeII site-directed mutagenesis kit (Agilent Technologies, USA) and primers listed in *Appendix 1—table 1*. To prepare lentiviral expression constructs, the ORF of FLAG-tagged SMS2 was PCR amplified using pcDNA3.1-FLAG-SMS2 as a template. The amplified DNA was inserted into pENTR11 (Invitrogen, A10467) using the BamHI and NotI restriction sites. Pathogenic mutations and/or mutations affecting active site residue Asp276 were introduced by site-directed mutagenesis as described above. The inserts of the pENTR11 constructs were transferred into the lentiviral expression vector pInducer20 (Addgene, 44012) using Gateway cloning (Invitrogen) according to the manufacturer's instructions. The constructs encoding FLAG-tagged EqtSM (pET28a-EQ-SM-3xFLAG), GFP-tagged EqtSM$_{SS}$ (pN1-EqtSM$_{SS}$-oxGFP) and GFP-tagged EqtSol$_{SS}$ (pN1-EqtSol$_{SS}$-oxGFP) were described in *Deng et al., 2016*. The constructs encoding GFP-tagged EqtSM$_{cyto}$ (pN1-EqtSM$_{cyto}$-oxGFP), GFP-tagged EqtSol$_{cyto}$ (pN1-EqtSol$_{cyto}$-oxGFP), and mKate-tagged EqtSM$_{cyto}$ (pN1-EqtSM$_{cyto}$-mKate) were described in *Niekamp et al., 2022*. The constructs encoding cytosolic GFP-tagged lysenin (LysSM$_{cyto}$) and a non-SM binding lysenin variant (LysSol$_{cyto}$) were kindly provided by Felix Randow (MRC-LMB, Cambridge, UK) and described in *Ellison et al., 2020*. The construct encoding mCherry-tagged VAPA was described in *Jain et al., 2017*. The construct encoding GFP-tagged Sec16L (pEFP-C1-Sec16L) was a kind gift from Benjamin Glick (University of Chicago, USA) and described in *Bhattacharyya and Glick, 2007*. The expression construct encoding mCherry-tagged D4H (pN1-D4H-mCherry) was kindly provided by Gregory Fairn (University of Toronto, Canada) and described in *Maekawa and Fairn, 2015*.

## Mammalian cell culture and transfection

Human fibroblasts derived from skin biopsies of OP-CDL patients and healthy controls were previously described in *Pekkinen et al., 2019*. Human fibroblasts, human cervical carcinoma HeLa cells (ATCC CCL-2), human osteosarcoma epithelial U2OS cells (ATCC HTB-96), and human embryonic kidney 293 cells transformed with Simian Virus 40 large T antigen (HEK293T, ATCC CRL-3216) were cultured in high-glucose Dulbecco's modified Eagle's medium (DMEM) containing 2 mM L-glutamine and 10% FBS, unless indicated otherwise. A HeLa cell-line lacking SMS1 and SMS2 (ΔSMS1/2) was described in *Niekamp et al., 2022*. Cell lines were routinely examined for their morphology and analyzed for their characteristic protein expression profiles. All cell-lines were free of mycoplasma contaminations as determined routinely by DAPI staining or PCR assay. DNA transfections were carried out using Lipofectamine 3000 (Thermo Fisher Scientific, L3000001) according to the manufacturer's instructions.

## Lentiviral transduction

HeLa ΔSMS1/2 cells were stably transduced with pInducer20 constructs encoding FLAG-tagged SMS2, SMS2$^{D276A}$, SMS2$^{I62S}$, SMS2$^{I62S/D276A}$, SMS2$^{M64R}$ or SMS2$^{M64R/D276A}$. To this end, low passage HEK293T cells were co-transfected with the corresponding pInducer20-FLAG-SMS2 construct and the packaging vectors psPAX2 (Addgene, 12260) and pMD2.G (Addgene, 12259). Culture medium was changed

6 hr post-transfection. After 48 hr, the lentivirus-containing medium was harvested, passed through a 0.45 µm filter, mixed 1:1 (v/v) with DMEM containing 8 µg/ml polybrene and used to infect HeLa ΔSMS1/2 cells. At 24 hr post-infection, the medium was replaced with DMEM containing 1 mg/ml G418 and selective medium was changed daily. After 5 days, positively transduced cells were analyzed for doxycycline-dependent expression of the FLAG-tagged SMS2 variant using immunoblot analysis, immunofluorescence microscopy and metabolic labeling with clickSph, as described below.

## Cell lysis and immunoblot analysis

Cells were harvested and lysed in Lysis Buffer (1% TritonX-100, 1 mM EDTA pH 8.0, 150 mM NaCl, 20 mM Tris pH 7.5) supplemented with Protease Inhibitor Cocktail (PIC; 1 µg/ml aprotinin, 1 µg/ml leupeptin, 1 µg/ml pepstatin, 5 µg/ml antipain, 157 µg/ml benzamidine). Nuclei were removed by centrifugation at 600 x g for 10 min at 4 °C. Post nuclear supernatants were collected and stored at –80 °C until use. Protein samples were mixed with 2 x Laemmli Sample Buffer (0.3 M Tris HCl, pH 6.8, 10% SDS, 50% glycerol, 10% 2-β-mercaptoethanol, 0.025% bromphenol blue), resolved by SDS-PAGE using 12% acrylamide gels, and transferred onto nitrocellulose membrane (0.45 µm; GE Health Sciences USA). Membranes were blocked with 5% non-fat milk solution for 40 min and washed with 0.05% Tween in PBS (PBST). Next, membrane was incubated for 2 hr with primary antibody in PBST, washed three times with PBST and incubated with HRP-conjugated secondary antibody in PBST. After washing in PBST, the membranes were developed using enhanced chemiluminescence substrate (ECL; Thermo Fisher Scientific, USA). Images were recorded using a ChemiDoc XRS +System (Bio-Rad, USA) and processed with Image Lab Software (BioRad, USA).

## Metabolic labelling and TLC analysis

Cells were metabolically labeled for 24 hr with 4 µM clickable sphingosine in Opti-MEM reduced serum medium without Phenol red (Gibco, 11058). Next, cells were washed with PBS, harvested, and subjected to Bligh and Dyer lipid extraction (*Bligh and Dyer, 1959*). Dried lipid films were click reacted in a 40 µl reaction mix containing 0.45 mM fluorogenic dye 3-azido-7-hydroxycoumarin (Jena Bioscience, CLK-FA047), 1.4 mM Cu(I)tetra(acetonitrile) tetrafluoroborate and 66% EtOH:CH-Cl$_3$:CH$_3$CN (66:19:16, v:v:v). Reaction mixtures were incubated at 40 °C for 4 hr followed by 12 hr incubation at 12 °C and applied at 120 nl/s to NANO-ADAMANT HP-TLC plates (Macherey-Nagel, Germany) with a CAMAG Linomat 5 TLC sampler (CAMAG, Switzerland). The TLC plate was developed in CHCl$_3$:MeOH:H$_2$O:AcOH (65:25:4:1, v:v:v:v) using a CAMAG ADC2 automatic TLC developer (CAMAG, Switzerland). The coumarin-derivatized lipids were visualized using a ChemiDoc XRS +with UV-transillumination and Image Lab Software (BioRad, USA).

## Fluorescence microscopy

For immunofluorescence microscopy, cells were grown on glass coverslips and fixed in 4% paraformaldehyde (PFA) for 15 min at RT. After quenching in 50 mM ammonium chloride, cells were permeabilized with permeabilization buffer (PBS containing 0.3% (v/v) Triton-X100 and 1% (v/v) BSA) for 15 min. Immunostaining was performed in permeabilization buffer and nuclei were counterstained with DAPI, as described in *Jain et al., 2017*. Coverslips were mounted onto glass slides using ProLong Gold Antifade Reagent (Thermo Fisher Scientific, USA). Fluorescence images were captured using a DeltaVision Elite Imaging System (GE Health Sciences, USA) or Leica DM5500B microscope (Leica, Germany), as indicated.

Imaging of live cells expressing SM or cholesterol reporters was performed using a Zeiss Cell Observer Spinning Disc Confocal Microscope equipped with a TempModule S1 temperature control unit, a Yokogawa Spinning Disc CSU-X1a 5000 Unit, a Evolve EMCDD camera (Photonics, Tucson), a motorized xyz-stage PZ-2000 XYZ (Applied Scientific Instrumentation) and an Alpha Plan-Apochromat x 63 (NA 1.46) oil immersion objective. The following filter combinations were used: blue emission with BP 445/50, green emission with BP 525/50, orange emission BP 605/70. All images were acquired using Zeiss Zen 2012 acquisition software. For hypotonic swelling, U2OS cells or fibroblasts were seeded in a µ-Slide 8 well glass bottom chamber (Ibidi; 80827) and transfected with indicated expression constructs. At 16 hr post-transfection, cells were imaged in isotonic medium (100% Opti-MEM) or after 5 min incubation in hypotonic medium (1% Opti-MEM in H$_2$0) at 37 °C. For cholesterol localization experiments, cells transfected with D4H-mCherry were incubated for 16 hr in growth medium

containing 70 µg/ml 10 kDa dextran conjugated with Alexa Fluor 647 (Thermo Fisher Scientific, D22914). Growth medium was replaced with Opti-MEM 2 hr prior to imaging. Images were deconvoluted using Huygens deconvolution (SVI, The Netherlands) and processed using Fiji software (NIH, USA).

Ratiometric confocal imaging of cells stained with Nile Red-based solvatochromic probes (NR12A, $NRER_{Cl}$) was performed on a Zeiss LSM 880 with an AiryScan module using a 63X1.4 NA oil immersion objective. Excitation was provided by a 532 nm laser and fluorescence emission was detected at two spectral ranges: 500–600 ($I500-600$) and 600–700 nm ($I600-700$). The images were processed with a home-made program under LabView, which generates a ratiometric image by dividing the image of the $I500-600$ channel by that of the $I600-700$ channel, as described in *Darwich et al., 2013*. For each pixel, a pseudo-color scale was used for coding the ratio, while the intensity was defined by the integral intensity recorded for both channels at the corresponding pixel. The computer code for processing the ratiometric images was kindly provided by Romain Vauchelles (Université de Strasbourg, France) and is available online: https://piq.unistra.fr/websites/pharmacie/PIQ/images/plugins_ImageJ/RatioIoJ.zip. For staining with the NR12A probe, cells were washed with PBS and incubated in Opti-MEM containing 0.2 µM NR12A for 10 min at RT. For staining with the $NRER_{Cl}$ probe, cells were washed with PBS and incubated in Opti-MEM containing 0.2 µM $NRER_{Cl}$ for 30 min at 37 °C. Subsequently, cells were washed 3 times with PBS and imaged in Opti-MEM.

## Cytotoxicity assay

Cells were seeded in 96-well plate (Greiner Bio-One; 655101) at 10,000 cells per well in DMEM supplemented with 10% FBS. After 24 hr, the medium was replaced with Opti-MEM, and 24 hr later mβCD was added at the indicated concentrations. After 1 hr, PrestoBlue HS (Thermo Fisher Scientific; P50200) was added to the well to a final concentration of 10% (v/v) and incubated for 3.5 hr at 37 °C. Next, absorbance at 570 nm was measured with 600 nm as reference wavelength using an Infinite 200 Pro M-Plex plate reader (Tecan Lifesciences). Measurements were average of technical quadruplicates. To calculate relative percentage of cell survival, the measured value for each well (x) was subtracted by the minimum measured value (min) and divided by the subtrahend of the average measured value of untreated cells (untreated) and the minimum measured value (min); ((x-min)/(untreated-min) *100).

## Cell surface staining with EqtSM

For production of recombinant FLAG-tagged EqtSM, *E. coli* BL21 (DE3) transformed with pET28a-EQ-SM-3xFLAG was grown at 37 °C to early exponential phase in LB medium containing 100 µg/ml ampicillin prior to addition of 0.4 mM isopropyl β-D-1-thiogalactopyranoside. After 5 hr induction, bacteria were mechanically lysed in 20 mM $Na_2HPO_4/NaH_2PO_4$, pH 7.4, 500 mM NaCl, and 25 mM imidazole supplemented with PIC by microtip sonication. Bacterial lysates were cleared by centrifugation at 10,000 *g* for 20 min at 4 °C and applied to a HisTrap HP column using an AKTA Prime protein purification system (GE Healthcare, Life Sciences). Bound protein was eluted with a linear imidazole gradient. HeLa cells grown on glass coverslips were incubated with 1 µM of purified FLAG-EqtSM in Labeling Buffer (20 mM $Na_2HPO_4/NaH_2PO_4$, pH 7.4, 500 mM NaCl) for 2 min at RT, washed with PBS, and then fixed in 4% PFA for 15 min at RT. After quenching in 50 mM ammonium chloride, cells were immunostained with polyclonal rabbit anti-FLAG antibody at 4 °C for 10 min and Cy2-conjugated anti-rabbit antibody at RT for 30 min in PBS supplemented with 1% BSA. For flow cytometry, HeLa cells were trypsinized, resuspended in medium containing 10% FCS, washed and then incubated with 1 µM Eqt-FLAG at RT for 2 min in Labeling Buffer. Next, cells were incubated with anti-FLAG antibody for 10 min at 4 °C in PBS containing 1% BSA, washed, and then fixed in 4% PFA for 15 min at RT. After quenching in 50 mM ammonium chloride, cells were incubated with Cy2-conjugated anti-rabbit antibody for 45 min in PBS containing 1% BSA, washed, and subjected to flow cytometry using a SH800 Cell Sorter (Sony Biotechnology). Flow cytometry data were analysed using Sony Cell Sorter software version 2.1.5.

## LC-MS/MS lipidomics

Cells were incubated for 24 hr in Opti-MEM reduced serum medium in the absence or presence of 1 µg/ml doxycycline. Next, cells were harvested in homogenization buffer (15 mM KCl, 5 mM NaCl, 20 mM HEPES/KOH pH 7.2, 10% glycerol, 1 x PIC) using a sonifier BRANSON 250.The protein in

crude homogenates was determined by Bradford protein assay (BioRad, USA) and 50 µg of protein was used for a subsequent chloroform/methanol extraction. To normalize lipid concentration of lipids in the samples, homogenates were prior to the extraction spiked with lipid standards ceramide (Cer 18:1/17:0) and sphingomyelin (SM 18:1/17:0). Dried lipid extracts were dissolved in a 50:50 mixture of mobile phase A (60:40 water/acetonitrile, including 10 mM ammonium formate and 0.1% formic acid) and mobile phase B (88:10:2 2-propanol/acetonitrile/$H_2O$, including 2 mM ammonium formate and 0.02% formic acid). HPLC analysis was performed on a C30 reverse-phase column (Thermo Acclaim C30, 2.1×250 mm, 3 µm, operated at 50 °C; Thermo Fisher Scientific) connected to an HP 1100 series HPLC system(Agilent) and a QExactive*PLUS* Orbitrap mass spectrometer (Thermo Fisher Scientific) equipped with a heated electrospray ionization (HESI) probe. MS analysis was performed as described previously (*Eising et al., 2019*). Briefly, elution was performed with a gradient of 45 min; during the first 3 min, elution started with 40% of phase B and increased to 100% in a linear gradient over 23 min 100% of B was maintained for 3 min. Afterwards, solvent B was decreased to 40% and maintained for another 15 min for column re-equilibration. MS spectra of lipids were acquired in full-scan/data-dependent MS2 mode. The maximum injection time for full scans was 100ms, with a target value of 3,000,000 at a resolution of 70,000 at m/z 200 and a mass range of 200–2000 m/z in both positive and negative mode. The 10 most intense ions from the survey scan were selected and fragmented with HCD with a normalized collision energy of 30. Target values for MS/MS were set at 100,000 with a maximum injection time of 50ms at a resolution of 17,500 at m/z of 200. Peaks were analyzed using the Lipid Search algorithm (MKI, Tokyo, Japan). Peaks were defined through raw files, product ion and precursor ion accurate masses. Candidate molecular species were identified by database (>1,000,000 entries) search of positive (+$H^+$; +$NH_4^+$) or negative ion adducts (-$H^-$;+$COOH^-$). Mass tolerance was set to 5 ppm for the precursor mass. Samples were aligned within a time window and results combined in a single report. From the intensities of lipid standards and lipid classes used, absolute values for each lipid in pmol/mg protein were calculated. Data are reported as mol% of total phospholipids measured.

## Organellar purification and shot-gun lipid mass spectrometry

### Affinity purification

A total of 10 million cells were seeded in 15 cm dishes and cultured for 24 hr in Opti-MEM reduced serum medium containing 1 µg/ml doxycycline. For ER purifications, cells were washed once in ice-cold PBS. For PM purification, cells were washed three times with ice-cold PBS and then incubated for 30 min at 4 °C in PBS containing 1 mg/ml EZ-Link-sulfo-NHS-LC-LC-Biotin (Thermo Fisher Scientific). Next, cells were washed three times with ice-cold PBS. Biotin-treated and untreated cells were scraped in ice-cold PBS, centrifuged once at 500 x g, and then twice at 1,000 x g for 5 min, 4 °C. All steps from here were performed on ice or at 4 °C. Cell pellets were re-suspended in ice-cold 10 ml SuMa buffer [10 mM Hepes, 0.21 M mannitol, 0.070 M Sucrose, pH 7.5]. After the third centrifugation step, cells were resuspended in 1 ml SuMa4 buffer [SuMa buffer supplemented with 0.5 mM DTT], 0.5% fatty acid-free BSA (Sigma Aldrich), 25 units/ml Benzonase (Sigma Aldrich) and 1 x cOmplete Mini, EDTA-free Protease Inhibitor Cocktail (Roche Diagnostics) and lysed by passages through a Balch homogenizer 20 times. The cell lysates were subsequently centrifuged at 1500 x g for 10 min and again for 15 min to prepare the light membrane fractions (LMFs). For ER purification, LMFs were incubated with rabbit anti-calnexin polyclonal antibody (Abcam) and subsequently with anti-rabbit IgG MicroBeads (Miltenyi Biotec), and for the PM purification, only with Streptavidin MicroBeads (Miltenyi Biotec). The LMFs were then loaded into MS Columns (Miltenyi Biotec) mounted on a magnetic stand (Miltenyi Biotec) and pre-equilibrated in SuMa4 buffer. The columns were washed three times with 500 µl SuMa4 buffer and twice with 500 µl SuMa2 buffer (SuMa4 without Benzonase and PIC). Thereafter, columns were removed from the magnetic stand and elution was performed with 600 µl SuMa2 buffer. Eluate samples were equally divided for western blotting and lipidomics. All samples were centrifuged at 21,100 x g for 20 min and supernatants were discarded. Pellets were resuspended in 200 µl SuMa +buffer (SuMa2 without BSA) and the centrifugation was repeated. Lipidomics samples were stored at –80 °C until analysis and immunoblot samples were dissolved in 2 x Laemmli Sample Buffer containing 100 mM DTT and stored at –20 °C until processing.

## Lipid extraction

Lipid extraction was performed as previously described (*Nielsen et al., 2020*), with some modifications. Briefly, samples in 200 µl 155 mM ammonium bicarbonate were mixed with 24 µl internal lipid standard mix (*Nielsen et al., 2020*) and 976 µl chloroform:methanol 2:1 (*v/v*). The samples were shaken in a thermomixer at 2000 rpm and 4 °C for 15 min and centrifuged for 2 min at 2000 x g and 4 °C. Then, the lower phase containing lipids was washed twice with 100 µl methanol and 50 µl 155 mM ammonium bicarbonate. Lower phase was then transferred to new tubes and dried in a vacuum centrifuge for 75 min, and the dried lipids were resuspended in 100 µl chloroform:methanol 1:2 (*v:v*).

## Mass spectrometry

Shotgun lipidomics was performed as previously described (*Nielsen et al., 2020*). Lipid extracts (10 µl) were loaded in a 96 well plate and mixed with either 12.9 µl-positive ionization solvent (13.3 mM ammonium acetate in propan-2-ol) or 10 µl-negative ionization solvent (0.2% (*v/v*) triethyl amine in chloroform:methanol 1:5 (*v/v*)). The samples were analyzed in the negative and positive ionization modes using Q Exactive Hybrid Quadrupole-Orbitrap mass spectrometer (Thermo Fisher Scientific, Waltham, MA) coupled to TriVersa NanoMate (Advion Biosciences, Ithaca, NY, USA). Data are reported as mol% of total lipids measured.

## Acknowledgements

We gratefully acknowledge Gregory Fairn, Benjamin Glick, Felix Randow and Christoph Thiele for providing DNA constructs and chemicals, Florian Fröhlich and Stefan Walter for technical assistance with LC-MS/MS, Rainer Kurre for technical assistance with live cell microscopy, and Romain Vauchelles for the computer code to process ratiometric images. We thank the Lipidomics Core Facility at the Danish Cancer Society Research Center for providing access to instruments and materials.

## Additional information

### Funding

| Funder | Grant reference number | Author |
|---|---|---|
| Deutsche Forschungsgemeinschaft | SFB944-P14 | Joost CM Holthuis |
| Deutsche Forschungsgemeinschaft | HO3539/1-1 | Joost CM Holthuis |
| Deutsche Forschungsgemeinschaft | SFB944-P8 | Jacob Piehler |
| National Institutes of Health | R35 GM144096 | Christopher G Burd |
| Novo Nordisk Foundation | NNF17OC0029432 | Kenji Maeda |
| Independent Research Fund Denmark | 6108–00542B | Kenji Maeda |

The funders had no role in study design, data collection and interpretation, or the decision to submit the work for publication.

### Author contributions

Tolulope Sokoya, Jan Parolek, Mads Møller Foged, Manuel Bozan, Bingshati Sarkar, Angelika Hilderink, Michael Philippi, Formal analysis, Validation, Investigation, Visualization, Methodology, Writing - review and editing; Dmytro I Danylchuk, Lorenzo D Botto, Paulien A Terhal, Outi Mäkitie, Jacob Piehler, Yeongho Kim, Christopher G Burd, Andrey S Klymchenko, Resources, Writing - review and editing; Kenji Maeda, Formal analysis, Supervision, Validation, Investigation, Visualization, Methodology,

Writing - review and editing; Joost CM Holthuis, Conceptualization, Supervision, Funding acquisition, Writing – original draft

## Author ORCIDs
Manuel Bozan  http://orcid.org/0000-0001-8426-369X
Outi Mäkitie  http://orcid.org/0000-0002-4547-001X
Jacob Piehler  http://orcid.org/0000-0002-2143-2270
Yeongho Kim  http://orcid.org/0000-0002-1477-925X
Christopher G Burd  http://orcid.org/0000-0003-1831-8706
Joost CM Holthuis  http://orcid.org/0000-0001-8912-1586

## Decision letter and Author response
Decision letter https://doi.org/10.7554/eLife.79278.sa1
Author response https://doi.org/10.7554/eLife.79278.sa2

---

## Additional files

### Supplementary files
• Appendix 1—figure 3—source data 1. Raw data of the quantitative analysis of lipid species under the conditions indicated.

• Transparent reporting form

### Data availability
All data generated or analysed during this study are included in the manuscript and supporting file. Source Data files have been provided for Figures 2-4, 7-9 and Appendix 1 - figure 3.

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

## Appendix 1

**Appendix 1—table 1.** Primers used for site-directed mutagenesis of SMS2.

| Primer name | Sequence |
| --- | --- |
| SMS2(I62S)-F | 5'-CGGACTATATCCAAAGTGCTATGCCCACTGAATC-3' |
| SMS2(I62S)-R | 5'-GATTCAGTGGGCATAGCACTTTGGATATAGTCCG-3' |
| SMS2(M64R)-F | 5'-CTATATCCAAATTGCTAGGCCCACTGAATCAAGG-3' |
| SMS2(M64R)-R | 5'-CCTTGATTCAGTGGGCCTAGCAATTTGGATATAG-3' |
| SMS2(D276A)-F | 5'-CGAACACTACACTATCGCTGTGATCATTGC-3' |
| SMS2(D276A)-R | 5'-GCAATGATCACAGCGATAGTGTAGTGTTCG-3' |

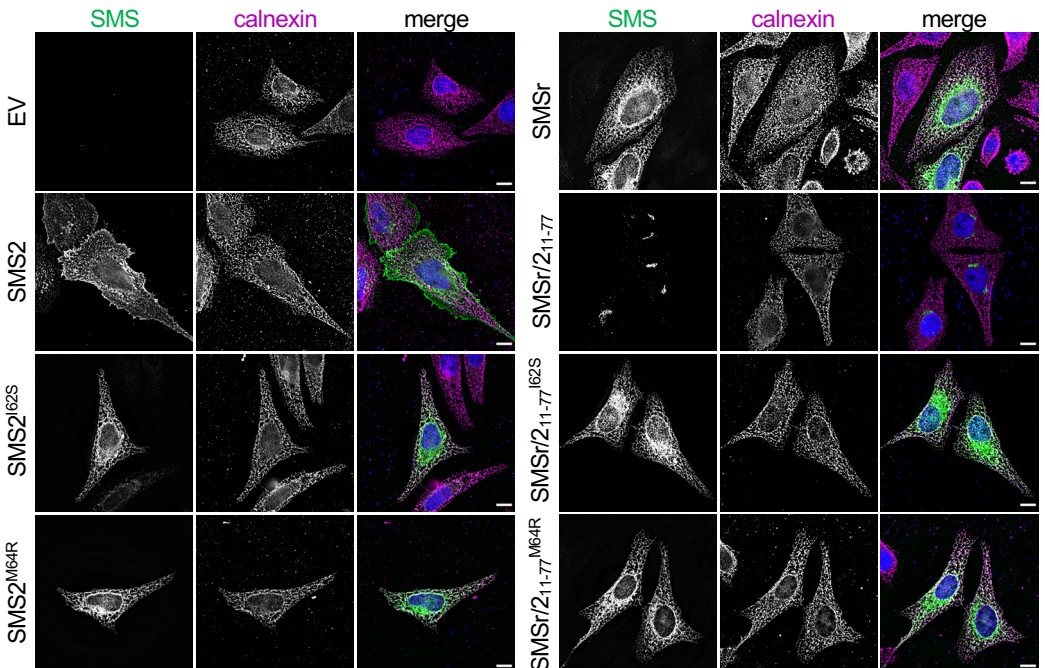

**Appendix 1—figure 1.** SMS2 contains an autonomous ER export signal. HeLa cells transfected with empty vector (EV) or FLAG-tagged SMS2, SMS2$^{I62S}$, SMS2$^{M64R}$, SMSr, SMSr/2$_{11-77}$, SMSr/2$_{11-77}$$^{I62S}$ or SMSr/2$_{11-77}$$^{M64R}$ were fixed, immunostained with a-FLAG (*green*) andα-calnexin (*magenta*) antibodies, counterstained with DAPI (*blue*) and imaged by DeltaVision microscopy. Scale bar, 10 μm.

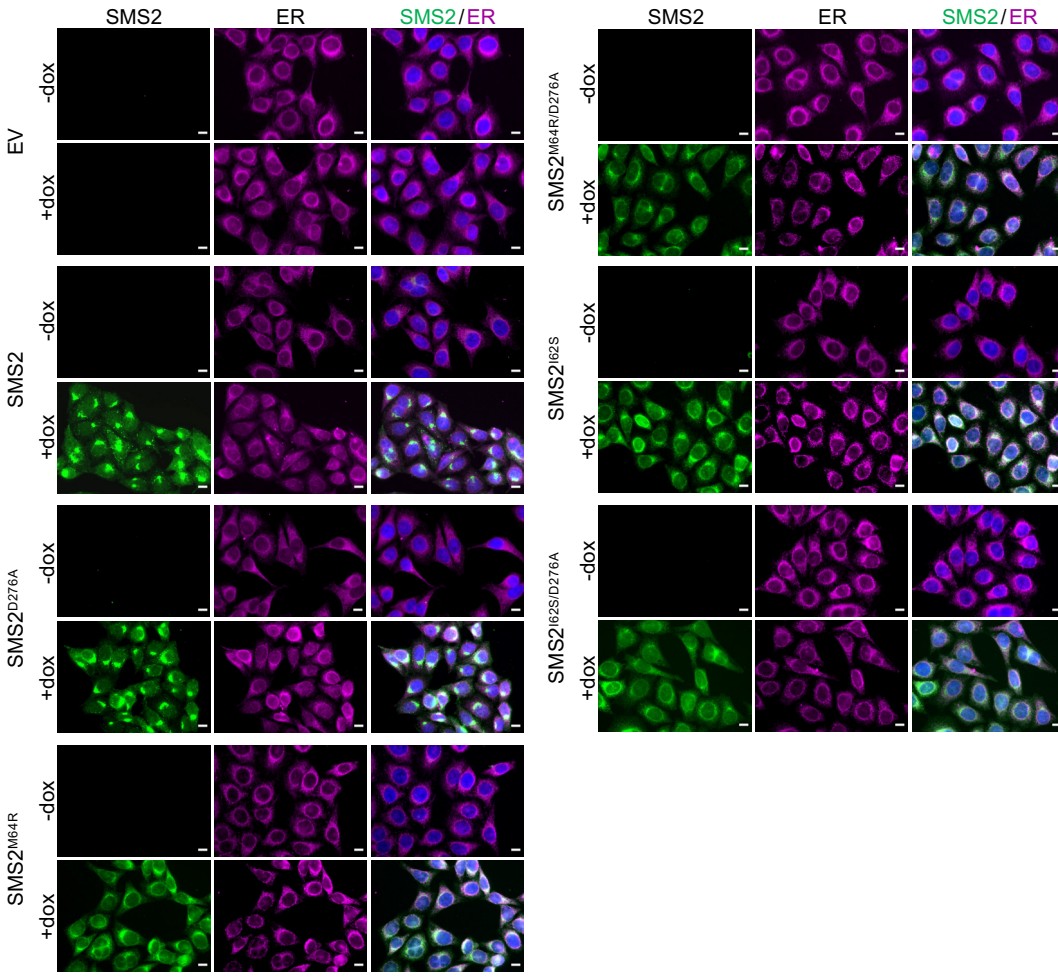

**Appendix 1—figure 2.** Doxycycline-induced expression of SMS2 variants in stably transduced HeLa cells. HeLa ΔSMS1/2 cells transduced with doxycycline-inducible Flag-tagged SMS2, SMS2$^{I62S}$, SMS2$^{M64R}$ or their enzyme-dead isoforms (D276A) were grown for 16 h in the absence or presence of 1 µg/ml doxycycline. Next, cells were fixed, immunostained withα-FLAG (*green*) and α-calnexin (*magenta*) antibodies, counterstained with DAPI (*blue*) and imaged by conventional fluorescence microscopy. Scale bar, 10 µm.

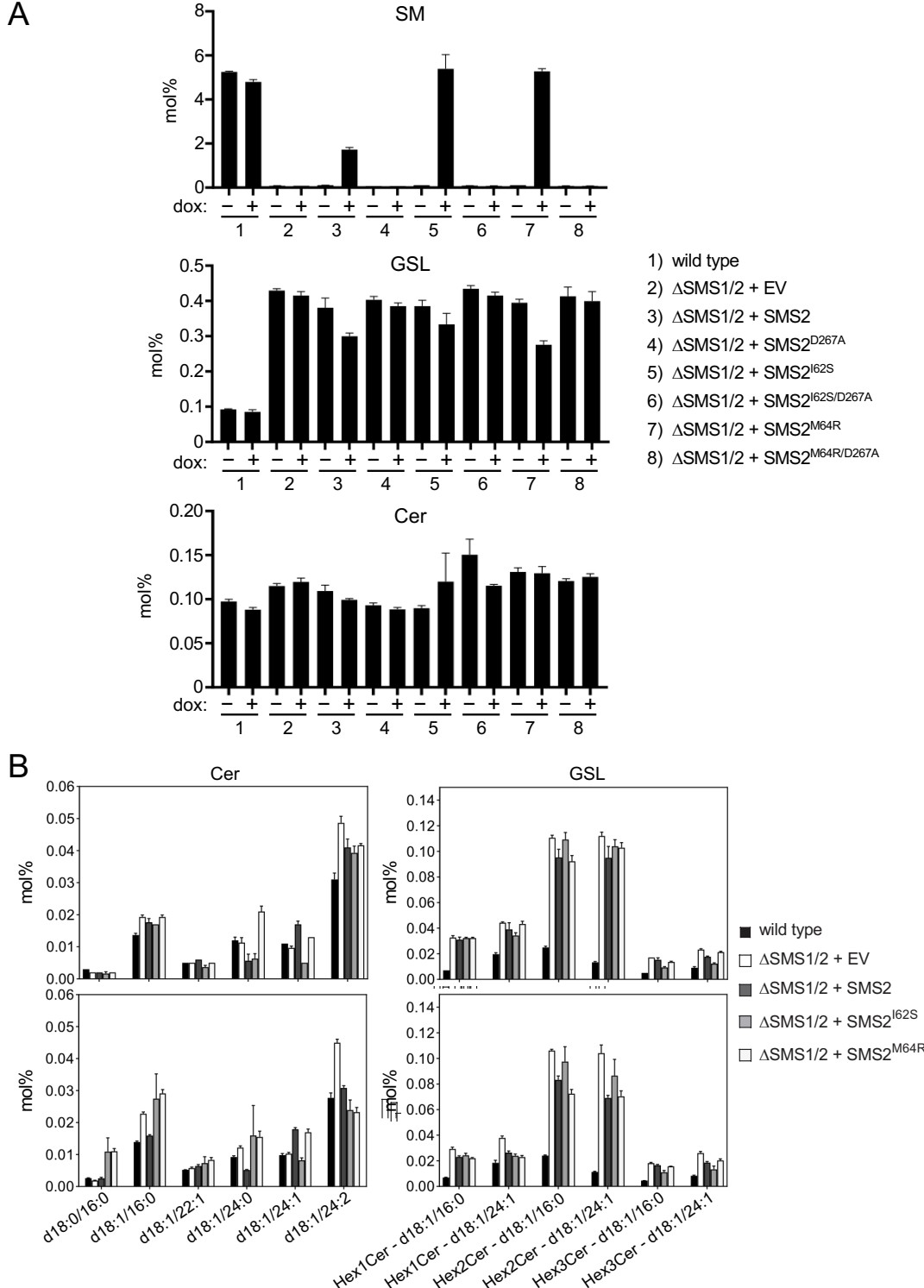

**Appendix 1—figure 3.** Pathogenic SMS2 variants support bulk production of SM in the ER. (**A**) HeLa ΔSMS1/2 cells transduced with doxycycline-inducible FLAG-tagged SMS2, SMS2^I62S, SMS2^M64R or their enzyme-dead isoforms (D276A) were grown for 16 h in the absence or presence of 1 µg/ml doxycycline and subjected to total lipid extraction. Cellular SM, glycosphingolipid (HexCer) and ceramide (Cer) levels were quantified by LC-MS/MS and expressed as mol% of total phospholipid analyzed. (**B**) Ceramide and HexCer species in total lipid extracts of cells treated as in (**A**) were quantified by LC-MS/MS and expressed as mol% of total phospholipid analyzed. All data are average ± SD of technical triplicates.

The online version of this article includes the following source data for appendix 1—figure 3:

**Appendix 1—figure 3—source data 1.** Raw data of the quantitative analysis of lipid species under the conditions indicated.

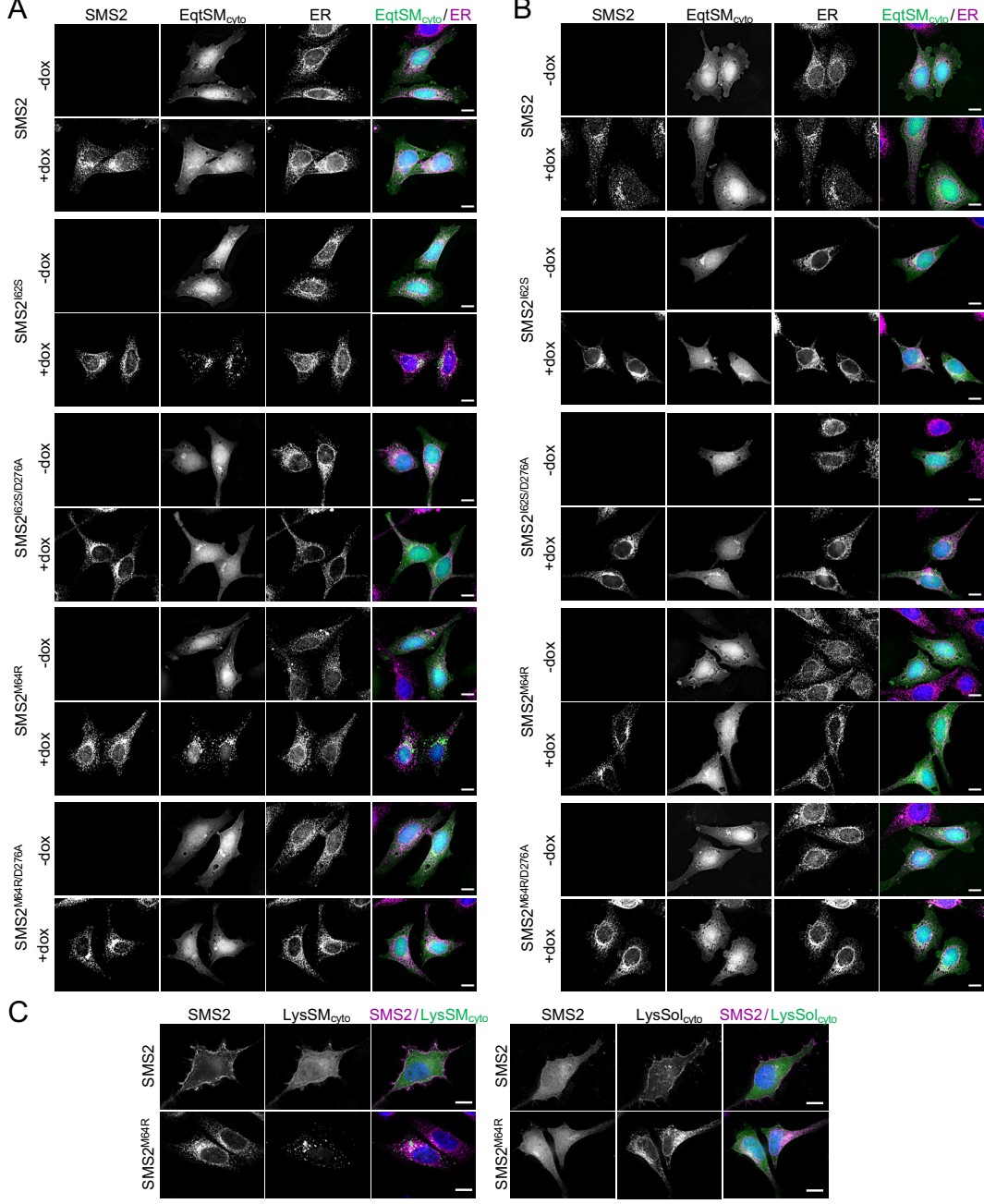

**Appendix 1—figure 4.** Mobilization of EqtSM$_{cyto}$ and LysSM$_{cyto}$ by pathogenic SMS2 variants. (**A**) HeLa ΔSMS1/2 cells transduced with FLAG-tagged SMS2, SMS2$^{I62S}$, SMS2$^{M64R}$ or their enzyme-dead isoforms (D276A) were transfected with GFP-tagged EqtSM$_{cyto}$ (*green*) and grown for 16 h in the absence or presence of 1 µg/ml doxycycline. Next, cells were fixed, immunostained withα-FLAG (SMS2, *red*) and α-calnexin (ER, *magenta*) antibodies, counterstained with DAPI (*blue*) and imaged by DeltaVision microscopy. (**B**) Cells as in (**A**) were transfected with GFP-tagged EqtSol$_{cyto}$ (*green*) and grown for 16 h in the absence or presence of 1 µg/ml doxycycline. Next, cells were fixed, immunostained withα-FLAG (SMS2, *red*) and α-calnexin (ER, *magenta*)
*Appendix 1—figure 4 continued on next page*

*Appendix 1—figure 4 continued*

antibodies, counterstained with DAPI (*blue*) and imaged by DeltaVision microscopy. (**C**) Cells as in (**A**) were transfected with GFP-tagged LysSM$_{cyto}$ or LysSol$_{cyto}$ (*green*) and grown for 16 h in the presence of 1 µg/ml doxycycline. Next, cells were fixed, immunostained withα-FLAG (SMS2, *magenta*), counterstained with DAPI (*blue*) and imaged by DeltaVision microscopy. Scale bar, 10 µm.

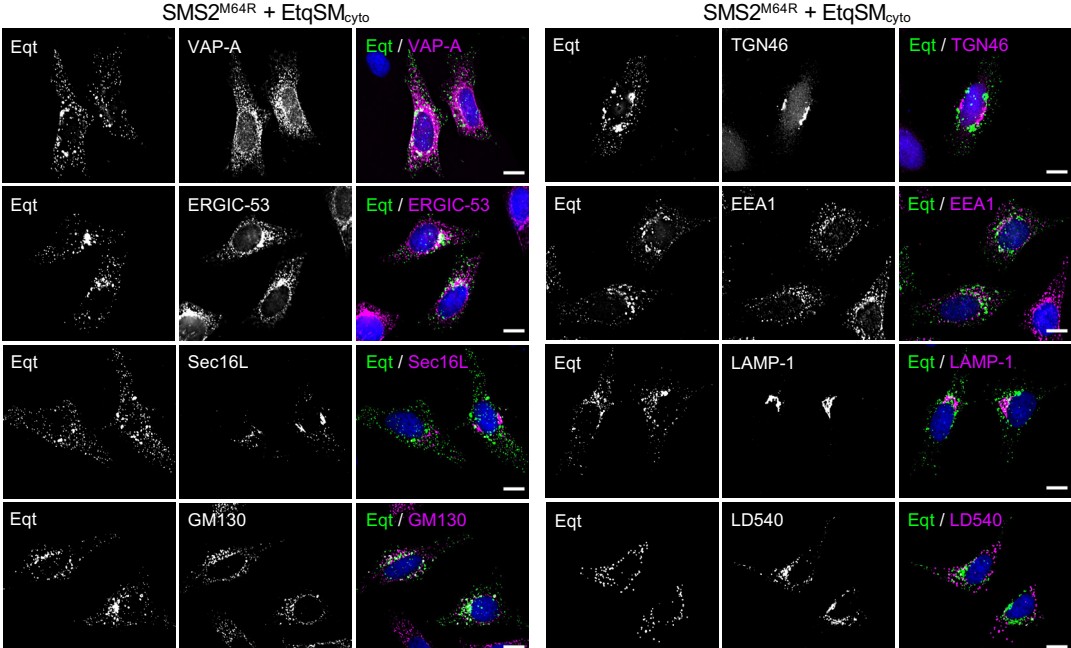

**Appendix 1—figure 5.** EqtSM$_{cyto}$-positive puncta formed in cell expressing pathogenic SMS2 variants do not co-localize with a wide range of organellar markers. HeLa ΔSMS1/2 cells transduced with doxycycline-inducible SMS2$^{M64R}$ were transfected with EqtSM$_{cyto}$ (*green*) and treated with 1 µg/ml doxycycline for 16 h. Next, cells were fixed, immunostained with antibodies against various organellar markers (*magenta*), counterstained with DAPI (*blue*) and imaged by DeltaVision microscopy. The ER was marked by co-transfection with mCherry-tagged VAPA while lipid droplets were labeled using the lipophilic dye LD540. Scale bar, 10 µm.

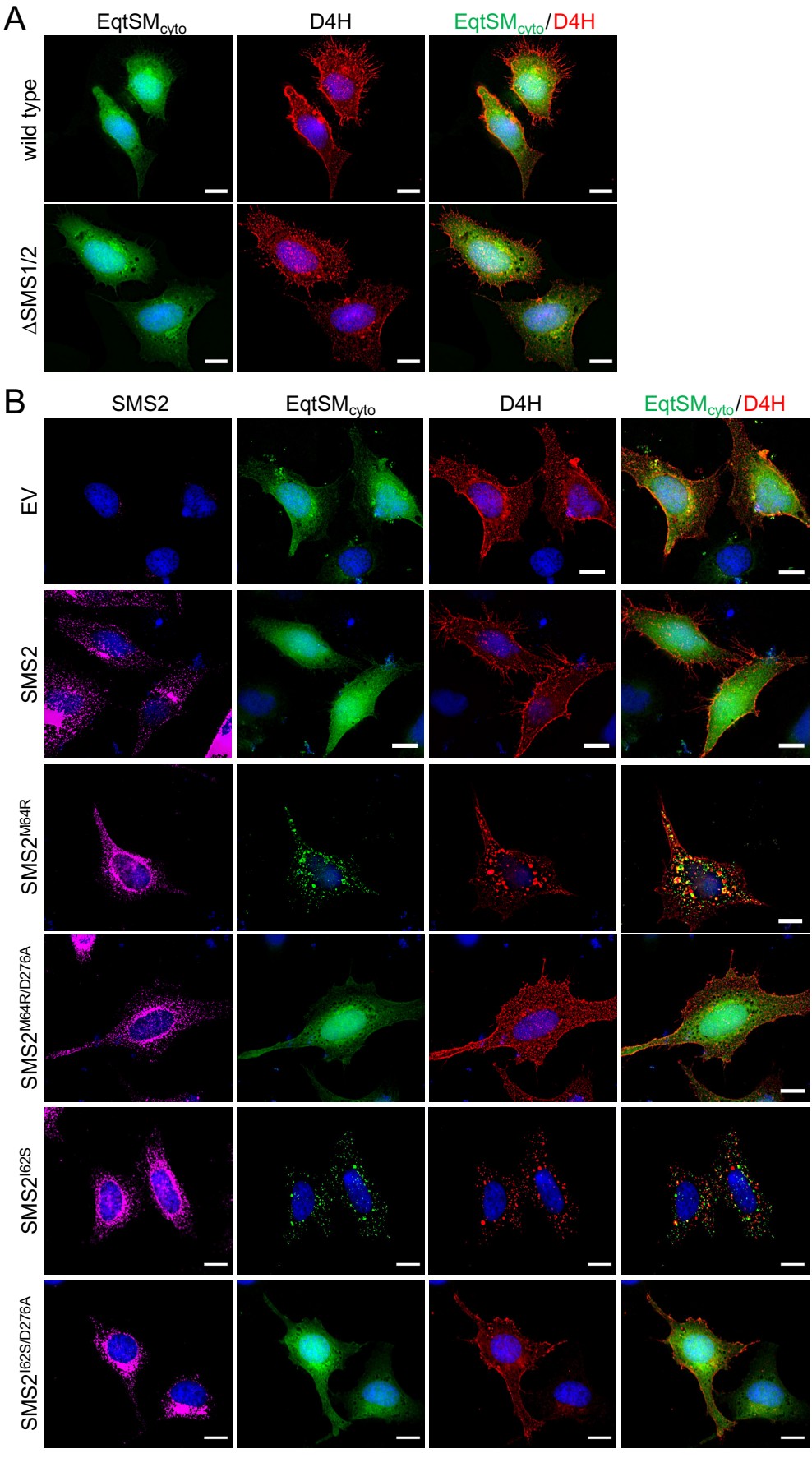

**Appendix 1—figure 6.** Pathogenic SMS2 variants perturb subcellular cholesterol pools. (**A**) HeLa wildtype orΔSMS1/2 cells were co-transfected with GFP-tagged EqtSM$_{cyto}$ (*green*) and mCherry-tagged D4H (*red*). After 16 h, cells were fixed, counterstained with DAPI (*blue*) and imaged by DeltaVision microscopy. (**B**) HeLaΔSMS1/2 cells transduced with FLAG-tagged SMS2, SMS2$^{I62S}$, SMS2$^{M64R}$ or their enzyme-dead isoforms (D276A) were co-transfected with GFP-tagged EqtSM$_{cyto}$ (*green*) and mCherry-tagged D4H (*red*) and then grown for 16 h in the presence of 1 µg/ml doxycycline. Next, cells were fixed, immunostained withα-FLAG antibodies (SMS2, *magenta*), counterstained with DAPI (*blue*) and imaged by DeltaVision microscopy. Scale bar, 10 µm.

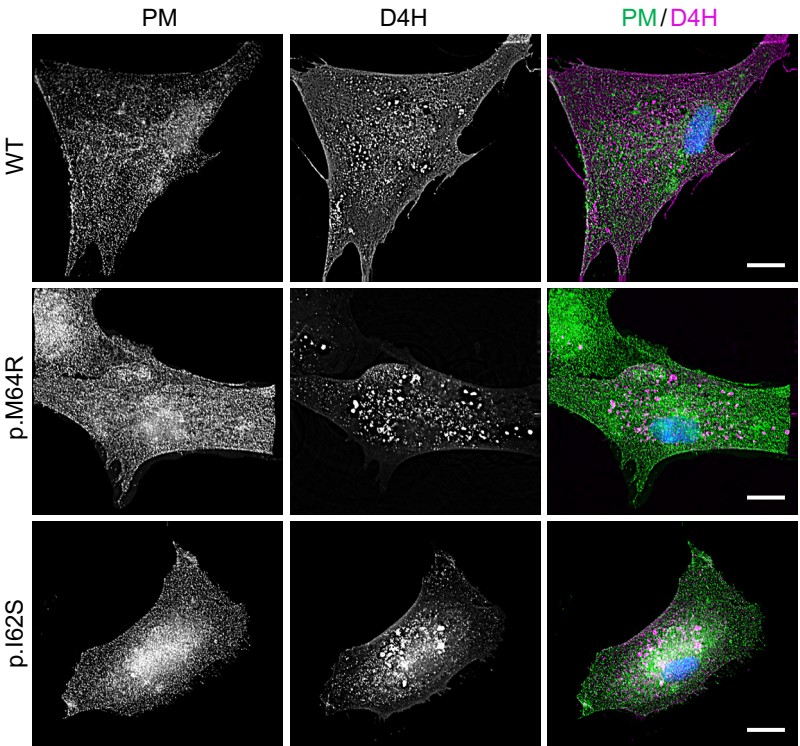

**Appendix 1—figure 7.** Patient-derived fibroblasts display perturbations in subcellular cholesterol pools. Control (WT) or patient-derived human skin fibroblasts carrying heterozygous missense variants c.185T>G (p.I62S) or c.191T>G (p.M64R) in *SGMS2* were transfected with mCherry-tagged D4H (*magenta*), fixed, immunostained withα-Na/K-ATPase antibodies (PM, *green*) and imaged by Deltavision microscopy. Scale bar, 10 µm.

