## [Editor Report]

Sphingomyelin synthase 2 (SMS2) is an enzyme located in the Golgi apparatus and the plasma membrane (PM) that mediates the synthesis of sphingomyelin (SM), a critical lipid in the PM. Mutations in SMS2 underlie a rare genetic disorder of bone formation. This useful study shows that the disease mutations cause retention of SMS2 in the ER, producing SM in the wrong place and leading to a disrupted SM/cholesterol gradient in the membranes of the secretory pathway. In addition to highlighting the roles of lipid gradients in cellular signaling pathways, this study also provides cell biologists with new tools to examine lipid localization in cells.

---

## [Decision Letter]

**Decision letter after peer review:**

Thank you for submitting your article "Pathogenic variants of sphingomyelin synthase SMS2 disrupt lipid landscapes in the secretory pathway" for consideration by *eLife*. Your article has been reviewed by 3 peer reviewers, and the evaluation has been overseen by a Reviewing Editor and Vivek Malhotra as the Senior Editor. The following individual involved in the review of your submission has agreed to reveal their identity: Anant K Menon (Reviewer #2).

Essential revisions:

Since the study hinges on accurate measurement of the leaflet distribution of sphingomyelin (SM), the authors should use an alternative to the cytosolic EqtSM reporter, such as lysenin as described by Reviewer #1 in point (a), to demonstrate the location of SM on the cytoplasmic leaflets of the PM and the ER to bolster their model. The authors should also tone down interpretations of some of their results, as indicated by the three reviewers below.

*Reviewer #1 (Recommendations for the authors):*

a) To detect cytosolic exposure of SM the authors use a mutated form of Equinatoxin (Eqt) fused to GFP. This mutant has been reported to show preferential binding to SM over PC, and to not induce membrane lysis. In the mutant cells, they find that this reporter translocates to cytosolic puncta which they are unable to identify as they do not co-localize with any organelle they test (although they do not test autophagosomes). This is not evidence that SM is on the cytoplasmic face of the ER or the PM. Indeed, it is not even clear that these structures have membranes in them, but whatever they are, they are clearly not the ER or the PM. One problem may be that the Eqt reporter used is perhaps not the most reliable reporter for SM, as its lipid preference has not been extensively characterized beyond the initial description. Thus, the authors should also use the different SM reporter, lysenin, which has been much more widely used than Eqt-based reporters and has been successfully applied to investigating cytosolic exposure of SM in living cells. A further issue is that their images of wild-type cells quite often show some concentration of Eqt-GFP in the perinuclear region (eg Figure 6A or 9B), although it is hard to see in some cases as the images show three different reporters simultaneously, with single channel images not provided. A similar perinuclear concentration has been seen with a different Eqt-based reporter in wild-type cells (Bakrac et al. JBC 285 22186 (2010)), and it may reflect Eqt not being an ideal reporter. Hopefully, a lysenin-based reporter will give clearer results.

b) The authors state that SMS2 has an ER export "IXMP motif". This is not a known motif, and indeed its sequence is not defined in this study as being IXMP. All they know is that mutations in the I and the M result in loss of ER export. They do not test the M, or indeed the other nearby resides that are also well conserved, and nor do they test sufficiency beyond transplanting a much larger part (residues 11-77) to an ER resident protein and showing it leaves the ER. Thus, all they can say is that part of SMS2 within residues 11-77 appears to be sufficient to confer ER export, and that mutation of I62 or M64 perturbs either an uncharacterized linear motif or the overall fold of the tail, such that it is no longer functional for ER export

c) When examining the location of proteins or reporters, the authors must show single channel images in gray scale of the key results (as they do in Figures 5 and 7). In several other figures, they only show multichannel images or color images, where the distribution is harder to see due to the other channels or being in color. An example is mentioned above for equinatoxin and is also the case in Figure 8, and some of the supplementary figures.

d) Do the levels of glycolipids fall in the mutant cells due to increased conversion of ceramide to SM?

e) The authors comment that cholesterol levels in the ER do not increase in the mutant. However, some caution may be needed as, firstly, cholesterol can partition out of membranes during fractionation, and secondly, cholesterol may not be able to reach the ER as the cells may not be making much in the ER but are instead obtaining it from lipoproteins in the culture medium.

f) The authors express EqtSM in the ER and report it being in mostly in the ER. This is surprising, as the paper describing this reporter showed that it is present throughout the secretory pathway (Deng et al. 2016). Moreover, this previous paper showed that the puncta that the authors comment on are actually post-Golgi carriers, and are not part of the ER as claimed here.

g) The organelle-specific reporters of lipid order are unlikely to be totally specific. The images shown in the paper reporting these reagents are of a low resolution and do not exclude multiple intracellular structures being labelled. Some note of caution is required here.

h) Why does over-expressing SMS2 not increase the levels of SM beyond that seen in wild-type cells?

i) The authors speculate that SM exposed to the cytosol could be cleaved by the neutral sphingomyelinase SMPD3 and this could be important for bone deposition. However, if this was the case then they would not be able to detect an accumulation of SM on cytosolic leaflets. Have they tried deleting SMPD3, or looking at the distribution of ceramide?

j) The authors should at least mention the puzzling finding that four patients with this bone-deposition disorder make a severely truncated version of SMS2 that contains only part of the N-terminal cytoplasmic tail.

*Reviewer #2 (Recommendations for the authors):*

I have several questions/comments.

1. Data: the increase in diunsaturated PC (Figure 3E, line 469) does not seem especially convincing, yet this is used as one of the main components of the overall description/conclusion that the lipid map of the cell is dysregulated. I wonder if the authors really want to comment on the ratio of di- versus mono-unsaturated PC – is this the point that they want to focus on? Perhaps this change in the ratio may be more dramatic than simply noting the change in di-unsaturated species.

2. Colours in Figure 3E are a bit confusing as they are the same as those used for the WC-ER pairing in panel D.

3. Data: Cholesterol levels are rather high in the whole cell sample (20 mole %) in Figures 3C and 4C – this should be <10% – is there any explanation? perhaps some aspect of the analysis is not being read out correctly? (this would have implications for values reported for the PM etc).

4. Data: Presumably the gels in Figures 3B and 4B are loaded on an equivalents basis (not protein basis) – this should be stated explicitly.

5. The accumulation of SM in the ER suggests the absence of an efficient vesicular or non-vesicular pathway to distribute this lipid to other compartments in sync with its rate of synthesis. Transport certainly occurs by both mechanisms as SM is found in both leaflets of the PM, but its rate seems not to be high enough. The discussion of this point in lines 438-447 is rather vague and it would be nice to see a slightly more quantitative discussion. The example of SM-exclusion from COPI vesicles pertains to lumenal SM. How do the authors envisage exclusion of the cytoplasm-facing pool?

6. Data: Line 271 and Figure 5 – the punctate versus uniform staining difference on hypotonic staining is not so clear – I wonder if this could be aided by a cartoon version of one of the images?

7. Typos: line 365 – consequently LOSE their viability; line 431 – prove not proof.

8. The discussion seems excessive – it should be reduced in length and sharpened to stay a little more focused on the actual data (there seem to be many extrapolations well beyond the scope of the data). Lines 463-465 are super-clear – they encapsulate key findings and could be used as a springboard for the discussion.

9. There are many beautifully executed experiments in this paper but I was hoping for an old-fashioned one that might indicate the disruption of the sterol/SM gradient – this would be the Brown-Rose experiment which monitored the trafficking of a GPI-anchored protein through the secretory pathway, showing that it became insoluble in cold TX-100 when it reached the Golgi apparatus. While the molecular basis of this phenomenon can be debated the experiment provides a clear readout of the lipid translation that the authors write about in the Introduction and Discussion. It's not required to do this, but the authors may want to consider if it would help develop their description of the phenotypes that they are studying.

10. If I am not mistaken, there are old data by Wieland and van Meer describing changes in transbilayer asymmetry of SM and GlcCer on BFA treatment – these data essentially report on the phenomenon of the symmetrization of the transbilayer distribution of SM at the ER. This work could be cited/discussed.

*Reviewer #3 (Recommendations for the authors):*

Sokoya, Parolek et al. impressively showed that pathogenic SMS2 variants result in a wide-ranging disruption of lipid distributions. Their careful mapping of organellar lipid changes is extremely interesting and valuable to the community. Complementing this with studying the effects on membrane order is helpful in trying to understand the overall consequences and, in the future, the pathobiochemistry of the disease. Certainly, this study will spawn multiple highly interesting avenues for research, some of which are elaborated on in the Discussion section. However, in the second part of the manuscript, a clear and testable hypothesis is missing. The authors mainly postulate that the production of SM at the ER dissipates the SM asymmetry not only at the ER but also at the PM. They show with several approaches that this might be the case, but some of which are complicated by technical difficulties that make a clear interpretation challenging. Details below.

The central claim – that SM production in the ER affects bilayer asymmetry in the PM – rests mainly on two observations: First, the lipidome analysis of PM isolates shows only a modest decrease in total SM levels in the disease variant, while the Eqt-staining of SM at the outer leaflet shows much reduced SM levels, leading the authors to conclude that SM is also present on the inner leaflet of the PM. Direct evidence of this would be strongly desired but was however complicated by the fact that the cytosolic EqtSM reporter could not detect SM at the inside of the PM. Likewise, the cytosolic EqtSM reporter did not directly detect SM on the cytosolic leaflet of the ER but instead localized to some yet unidentified punctate structures. As the nature of the punctae could not be clarified, an artefact induced by Eqt-SM binding (aggregates? condensates?) cannot be ruled out. Similar concerns apply to the use of EqtSMss in the ER lumen, which under isotonic show strange punctae (most of which cluster around the perinuclear region) that do not convincingly co-localize with calnexin. A clear binding of EqtSMss to calnexin-labeled membranes could only be shown in hypotonic conditions. In summary, while the use of biosensors for studying lipid localization is essential to this study, as the integrity of the cells must be preserved to properly study questions of lipid asymmetry, their use is accompanied by drawbacks that were not fully addressed. In particular, the reason why cytosolic EqtSM is not able to detect SM on the inner leaflet of the PM remains unclear.

The connection of SM mislocalization to subcellular cholesterol pools is very interesting, but not fully explored. Somewhat surprisingly, the authors found that ER cholesterol remains low even in conditions of high ER SM levels, and similarly, overall PM cholesterol levels are not affected in the disease model. Yet, detection of Chol at the inner leaflet of the PM by D4H is no longer possible when using SMS2-mutants. Instead, D4H localizes to endolysosomes. The authors do not comment on how this alternative localization potentially arises (increased Golgi-to-endosome trafficking? Cholesterol export defects of the lysosome in conditions of SM abundance?) nor confirm it with alternative methods (such as Filipin staining). Instead, they claim that the previously hypothesized SM pool in the inner leaflet of the PM renders the cholesterol inaccessible to the reporter. Here, again, potential artefacts of the reporter cannot be excluded and should be controlled for. For instance, investigating the cholesterol levels on the outer leaflet using recombinant reporters (as performed for SM) would help clarify if cholesterol distribution itself or indeed just their complex formation with SM is affected.

For better comprehension, the authors could provide a graphical representation of the downstream effects of SM production at the ER and its effects on cholesterol.

At several points in the manuscript, the authors use unpublished data to support their arguments. These data should be provided.

---

## [Author Response]

Reviewer #1 (Recommendations for the authors):a) To detect cytosolic exposure of SM the authors use a mutated form of Equinatoxin (Eqt) fused to GFP. This mutant has been reported to show preferential binding to SM over PC, and to not induce membrane lysis. In the mutant cells, they find that this reporter translocates to cytosolic puncta which they are unable to identify as they do not co-localize with any organelle they test (although they do not test autophagosomes). This is not evidence that SM is on the cytoplasmic face of the ER or the PM. Indeed, it is not even clear that these structures have membranes in them, but whatever they are, they are clearly not the ER or the PM.

We thank the reviewer for these valuable comments, which led us to perform additional experiments and modify the manuscript as outlined below.

Using two complementary approaches, we demonstrate that cells expressing pathogenic SMS2 variants contain bulk amounts of SM in the ER (Figures 3D and 5). Previous work established that the ER harbours low specificity scramblases that catalyse flip flop of phospholipids and sphingolipids with similar kinetics (Buton et al., 2002; Chalat et al., 2012). This led us to postulate that translocation of EqtSM_cyto_ to cytosolic puncta is driven by scrambling and cytosolic exposure of SM produced in the luminal ER leaflet. As pointed out by the reviewer, a puzzling aspect of this scenario is that the EqtSM_cyto_-positive puncta are largely segregated from ER markers (Figure 6A and Appendix – figure 5). We have now recapitulated this phenomenon using lysenin as alternative cytosolic SM reporter (see our response to comment #2 below). Moreover, we show that pathogenic SMS2 variants also trigger translocation of EqtSM_SS_ to luminal puncta that are continuous with the ER network (Figure 5A). New co-localization experiments revealed considerable overlap between the luminal and cytosolic puncta (new Figure 6D, upper row), suggesting that the latter originate at least in part from the ER. Upon hypotonic swelling, the luminal puncta become largely continuous with the ER membrane (Figure 5B) and tend to segregate from the cytosolic puncta (new Figure 6D, bottom row). Under these conditions, the cytosolic puncta give rise to small vesicular structures, implying the presence of a membranous core. As requested by Reviewer #2 (comment #6), we now included a cartoon to illustrate our findings (new Figure 6E).

Our novel findings reinforce the notion that SM produced by pathogenic SMS2 variants is scrambled across the ER bilayer, triggering mobilization of both luminal and cytosolic EqtSM. Whether the condensation of EqtSM into puncta reflects binding of the reporter to pre-existing SM microdomains or involves a clustering of SM-bound reporters driven by a residual pore-forming activity remains to be established. Resolving this issue will require a substantial amount of additional work. Conceivably, binding of EqtSM to the membrane may alter its curvature and even pinch off SM-enriched membrane regions, analogous to the shedding of extracellular vesicles observed when cells are exposed to the cholesterol/SM binding protein ostreolysin A (Skocaj et al., 2016; https://doi.org/10.1016/ j.bbamem.2016.08.015). In fact, such shedding may explain why EqtSM_cyto_-positive puncta do not co-localize with a wide range of organellar markers (Appendix – figure 5) and why externally added recombinant EqtSM preferentially labels the positively-curved edges of adherent cells (Figure 7A). While our data underline the value of EqtSM and LysSM as tools for probing SM topology in intact cells, we now added a note of caution regarding their potential impact on the structural organization of SM-rich membranes (p. 9, lines 317320).

One problem may be that the Eqt reporter used is perhaps not the most reliable reporter for SM, as its lipid preference has not been extensively characterized beyond the initial description. Thus, the authors should also use the different SM reporter, lysenin, which has been much more widely used than Eqt-based reporters and has been successfully applied to investigating cytosolic exposure of SM in living cells.

As referred to in the manuscript, we previously used EqtSM_cyto_ to demonstrate Ca^2+^-activated SM scrambling and exposure on the cytosolic surface of lysosomes damaged by bacterial pathogens or pore-forming drugs (Niekamp et al., 2022, Nature Commun 13, 1875). In that study, we demonstrate that mobilization of EqtSM_cyto_ to damaged lysosomes is abolished upon targeting a bacterial SMase to the cytosolic surface of the lysosomes. Nevertheless, as requested by the reviewer, we now used lysenin as alternative cytosolic SM reporter (LysSM_cyto_). As shown in new Figure 6B and C, cells expressing pathogenic SMS2 variants mobilize GFP-tagged LysSM_cyto_ to cytosolic puncta that colocalize extensively with mKate-tagged EqtSM_cyto_. These puncta do not form in cells expressing wildtype SMS2 or when using a non-SM binding variant of the probe (LysSol_cyto_). Our novel findings demonstrate that formation of cytosolic puncta in cells expressing pathogenic SMS2 variants is not an artefact inherent to the use of EqtSM.

A further issue is that their images of wild-type cells quite often show some concentration of Eqt-GFP in the perinuclear region (eg Figure 6A or 9B), although it is hard to see in some cases as the images show three different reporters simultaneously, with single channel images not provided. A similar perinuclear concentration has been seen with a different Eqt-based reporter in wild-type cells (Bakrac et al. JBC 285 22186 (2010)), and it may reflect Eqt not being an ideal reporter. Hopefully, a lysenin-based reporter will give clearer results.

We now provide single channel images of cells throughout our study. In single channel images of Figure 6A, which are presented in Appendix 1 – figure 4A and B, one can occasionally observe some concentration of GFP-tagged EqtSM in the perinuclear region of control cells. However, this type of staining can also be observed when using GFP-tagged EqtSol, a EqtSM variant that lacks affinity for SM and does not form puncta in cells expressing pathogenic SMS2 variants. Unlike EqtSM used in our study, the Eqt-based reporter used by Bakrac et al. (2010) was not engineered to reduce its cytotoxicity. Consequently, native Eqt retains the ability to form pores and may therefore more readily insert into membranes exposing small amounts of SM on the cytosolic surface, with the resulting damage potentially promoting further SM exposure and Eqt binding. Bakrac et al. (2010) found that native Eqt primarily stains the *cis-*Golgi, which may not be surprising given its close proximity to the site of bulk SM production and low content of cholesterol, a molecule known to reduce spontaneous flip-flop. We clearly do not see this phenomenon with EqtSM. Moreover, our finding that EqtSM and LysSM co-localize in cytosolic puncta that form only in cells expressing a catalytically active pathogenic SMS2 variant (Figure 6B and C) supports the notion that EqtSM is a reliable SM reporter.

b) The authors state that SMS2 has an ER export "IXMP motif". This is not a known motif, and indeed its sequence is not defined in this study as being IXMP. All they know is that mutations in the I and the M result in loss of ER export. They do not test the M, or indeed the other nearby resides that are also well conserved, and nor do they test sufficiency beyond transplanting a much larger part (residues 11-77) to an ER resident protein and showing it leaves the ER. Thus, all they can say is that part of SMS2 within residues 11-77 appears to be sufficient to confer ER export, and that mutation of I62 or M64 perturbs either an uncharacterized linear motif or the overall fold of the tail, such that it is no longer functional for ER export

In the manuscript we state that the IXMP motif in SMS2 is part of an autonomous ER export signal and do not describe it anywhere as ER export motif. As pointed out by the reviewer, and now also explicitly referred to in the manuscript (p. 5, lines 154-156), additional work will be required to determine whether mutation of Ile62 or Met64 perturbs an uncharacterized linear sequence motif or the overall fold of the enzyme’s *N*-terminal cytosolic tail, such that it is no longer functional for ER export.

c) When examining the location of proteins or reporters, the authors must show single channel images in gray scale of the key results (as they do in Figures 5 and 7). In several other figures, they only show multichannel images or color images, where the distribution is harder to see due to the other channels or being in color. An example is mentioned above for equinatoxin and is also the case in Figure 8, and some of the supplementary figures.

We now provide single images in gray scale for all key microscopy data. For Figure 1C and D, see Appendix – figure 1; for Figure 6A and B, see Appendix – figure 4A-C.

d) Do the levels of glycolipids fall in the mutant cells due to increased conversion of ceramide to SM?

We assume the reviewer refers to the LC-MS/MS data in Appendix 1 – figure 3, which represent SM and glycosphingolipid levels in wildtype and SMS1/2-DKO cells expressing active or enzyme-dead versions of SMS2 or its pathogenic variants. These data revealed a strong inverse correlation between cellular SM and glycosphingolipid (GSL) levels. As described on p. 5 (lines 172-176), this is consistent with a competition between Golgi-resident SM and glucosylceramide synthases for ceramide substrate.

e) The authors comment that cholesterol levels in the ER do not increase in the mutant. However, some caution may be needed as, firstly, cholesterol can partition out of membranes during fractionation, and secondly, cholesterol may not be able to reach the ER as the cells may not be making much in the ER but are instead obtaining it from lipoproteins in the culture medium.

We thank the reviewer for these valuable comments. If cholesterol were to partition out of membranes during isolation of the organelles, our lipidomics data indicate that this did not happen to an extent sufficient to wipe out the marked difference in cholesterol content between the ER and plasma membrane (~8 vs ~40 mol%, respectively; Figures 3D and 4D). In the revised manuscript, we modified the Discussion to acknowledge that cultured cells may not make much cholesterol in the ER but instead obtain bulk cholesterol from lipoproteins in the culture medium (p. 12, lines 444-447). We now also refer to previous work indicating that a stream of cholesterol constantly travels from the plasma membrane to the ER (Infante and Radhakrishnan, 2017). This enables the ER to continuously sample the cholesterol content of the plasma membrane and, if necessary, make adjustments via transcriptional regulation of cholesterol uptake and synthesis. Given the crucial role of SM in sequestering cholesterol (Slotte, 2013; Das et al., 2014), this raises the question how cells with plasma membrane-like levels of SM in the ER manage to keep their ER cholesterol levels low in the face of a continuous exchange of cholesterol between the plasma membrane and ER.

f) The authors express EqtSM in the ER and report it being in mostly in the ER. This is surprising, as the paper describing this reporter showed that it is present throughout the secretory pathway (Deng et al. 2016). Moreover, this previous paper showed that the puncta that the authors comment on are actually post-Golgi carriers, and are not part of the ER as claimed here.

When expressed in HeLa cells, we found the luminal reporter EqtSM_SS_ mainly concentrated in the perinuclear region and in punctate structures, consistent with the study of Deng et al. (2016). On the other hand, when expressed in U2OS cells or human skin fibroblasts, the reporter primarily localizes to the ER (see Author response image 1 and Figure 5A). Why EqtSM_SS_ is largely retained in the ER of the latter two cell types is not clear. Nevertheless, this allowed us to probe patient-derived fibroblasts for the presence of SM in the ER (Figure 9A).

**Author response image 1. sa2fig1:** 

g) The organelle-specific reporters of lipid order are unlikely to be totally specific. The images shown in the paper reporting these reagents are of a low resolution and do not exclude multiple intracellular structures being labelled. Some note of caution is required here.

The organelle-specific reporters of lipid order used in our study do not have absolute specificity, and we now raise this point in the revised manuscript as outlined below. Nevertheless, the article in which we describe these reporters for the first time includes high-resolution microscopy images of HeLa cells labelled with NR12A or NRER_Cl_ and stained for various organellar markers (Figure S3 in Danylchuk et al. 2021; https://pubs.acs.org/doi/pdf/10.1021/jacs.0c10972). The corresponding Pearson’s coefficients (listed in Table S2) show high correlation values for NR12A and NRER_Cl_ with plasma membrane and ER markers, respectively. Except for some minor staining of endosomes due to endocytosis, NR12A cannot really stain any other organelle than the plasma membrane as it cannot cross the plasma membrane. For NRERCl, some non-specific staining of organelles other than the ER cannot be excluded. Therefore, we now included the following sentence in the revised manuscript: “A propyl chloride group in NRER_Cl_ targets this probe to the ER (Danylchuck et al., 2021) even though some residual staining of other organelles cannot be excluded” (p. 10, lines 345-347).

h) Why does over-expressing SMS2 not increase the levels of SM beyond that seen in wild-type cells?

We previously showed that SMS1, not SMS2, is responsible for bulk SM production in HeLa cells (Tafesse et al., 2007; J. Biol. Chem. 282, 17537-17547). Unlike SMS1, which exclusively resides in the *trans*-Golgi, SMS2 populates both the *trans*-Golgi and plasma membrane. Thus, even when SMS2 is overexpressed, a substantial portion of the total enzyme pool is situated distal from SMS1 and Golgi-resident glucosylceramide synthase with respect to receiving ceramides synthesized in the ER. This may explain why SMS2 is less effective in restoring total cellular SM levels in comparison to pathogenic SMS2 upon their expression in SMS1/2 double KO HeLa cells (Figures 3D and 4D; Appendix 1 – figure 3A).

i) The authors speculate that SM exposed to the cytosol could be cleaved by the neutral sphingomyelinase SMPD3 and this could be important for bone deposition. However, if this was the case then they would not be able to detect an accumulation of SM on cytosolic leaflets. Have they tried deleting SMPD3, or looking at the distribution of ceramide?

We thank the reviewer for raising this valuable point. Whereas SMPD3-associated neutral SMase activity is abundantly present in osteosarcoma SaOS2 cells, we found that the enzyme is barely expressed in HeLa cells (Niekamp et al., 2022, Nature Commun 13, 1875) and skin fibroblasts (our unpublished data). This may explain why expression of pathogenic SMS2 variants in HeLa cells and skin fibroblasts, but not SaOS2 cells, triggers the mobilization of EqtSM_cyto_ (Figures 6A and 9B; our unpublished data). To challenge this possibility and gain further insight into the pathogenic mechanism underlying OP-CDL, we deleted SMPD3 in SaOS2 cells. As SMPD3 removal severely perturbed cell growth and survival, we generated a SaOS2 cell-line in which SMPD3 is under control of a doxycycline-inducible promoter. Experimental work to address the above issues using this cell-line is in progress.

j) The authors should at least mention the puzzling finding that four patients with this bone-deposition disorder make a severely truncated version of SMS2 that contains only part of the N-terminal cytoplasmic tail.

We assume the reviewer is referring to the nonsense pathogenic SMS2 variant p.Arg50*, which is more common in patients with OP-CDL and associated with a milder bone phenotype. We originally anticipated that this variant encodes a severely truncated and enzyme-dead version of SMS2 that contains only part of the *N*-terminal cytosolic tail (Pekinnen *et al.,* 2019). However, as discussed on p. 14, lines 498-509, our recent work revealed that Met64 in this variant is utilized as alternative start codon, yielding a truncated but catalytically active enzyme that is mislocalized to the *cis/medial* Golgi (T. Sokoya and J. Holthuis, unpublished data). Unlike pathogenic variants SMS2^I62S^ and SMS2^M64R^, SMS2^R50X^ would have no direct access to ER-derived ceramides and must compete with GlcCer synthase for ceramides delivered to the *cis/medial* Golgi. Backflow of SM produced in the *cis/medial* Golgi to the ER may affect the lipid landscape in the secretory pathway of cells expressing SMS2^R50X^, but to a lesser extent than that observed in cells expressing the pathogenic variants SMS2^I62S^ and SMS2^M64R^. Experiments to verify this idea are in progress and will be part of another manuscript that focuses entirely on a detailed functional characterization of the pathogenic SMS2^R50X^ variant.

Reviewer #2 (Recommendations for the authors):I have several questions/comments.1. Data: the increase in diunsaturated PC (Figure 3E, line 469) does not seem especially convincing, yet this is used as one of the main components of the overall description/conclusion that the lipid map of the cell is dysregulated. I wonder if the authors really want to comment on the ratio of di- versus mono-unsaturated PC – is this the point that they want to focus on? Perhaps this change in the ratio may be more dramatic than simply noting the change in di-unsaturated species.

Please note that in the Results section (p. 7, lines 232-235) we stated that the expression of SMS2^M64R^ is accompanied by an enhanced unsaturation of bulk phospholipid in the ER, as indicated by a significant rise in diunsaturated PC at the expense of saturated and mono-unsaturated PC species (Figure 3E). As this trend could be reproduced in 4 independent experiments, we felt it was appropriate to conclude that “The dramatic rise in ER-associated SM levels caused by pathogenic SMS2 variants was accompanied by a marked increase in PC desaturation” (Discussion, p. 13, lines 482-484).

2. Colours in Figure 3E are a bit confusing as they are the same as those used for the WC-ER pairing in panel D.

For consistency and to avoid confusion regarding pairing of data sets, we have now changed the colour coding in both Figures 3 and 4.

3. Data: Cholesterol levels are rather high in the whole cell sample (20 mole %) in Figures 3C and 4C – this should be <10% – is there any explanation? perhaps some aspect of the analysis is not being read out correctly? (this would have implications for values reported for the PM etc).

For HeLa cells we consistently measure a cholesterol content of ~20 mol% of total lipids. Moreover, the values we report here for the cholesterol content of isolated ER and plasma membrane (~8 and ~40 mol% of total lipids, resp.; Figures 3D and 4D) are in close agreement with those reported in other studies (Infante and Radhakrishnan, 2017, *eLife* e25466; doi: 10.1007/978-1-4939-9136-5_12).

4. Data: Presumably the gels in Figures 3B and 4B are loaded on an equivalents basis (not protein basis) – this should be stated explicitly.

We thank the reviewer for raising this point. As now explicitly indicated in the legends of Figures 3B and 4B, samples were loaded on equivalents basis.

5. The accumulation of SM in the ER suggests the absence of an efficient vesicular or non-vesicular pathway to distribute this lipid to other compartments in sync with its rate of synthesis. Transport certainly occurs by both mechanisms as SM is found in both leaflets of the PM, but its rate seems not to be high enough. The discussion of this point in lines 438-447 is rather vague and it would be nice to see a slightly more quantitative discussion. The example of SM-exclusion from COPI vesicles pertains to lumenal SM. How do the authors envisage exclusion of the cytoplasm-facing pool?

We thank the reviewer for pointing out that any curvature-based exclusion of SM from COPI vesicles would pertain to the luminal SM pool only. This led us to modify the corresponding paragraph in the Discussion and provide a more quantitative description of how pathogenic SMS2 variants undermine formation of a SM gradient along the secretory pathway (p. 13, lines 460-463).

6. Data: Line 271 and Figure 5 – the punctate versus uniform staining difference on hypotonic staining is not so clear – I wonder if this could be aided by a cartoon version of one of the images?

We thank the reviewer for this helpful suggestion. As requested, and in response to comment #a of Reviewer #1, we now included a cartoon to illustrate our findings when probing intracellular SM pools with luminal and cytosolic versions of EqtSM under isotonic and hypotonic conditions (Figure 6E).

7. Typos: line 365 – consequently LOSE their viability; line 431 – prove not proof.

These typos are now corrected.

8. The discussion seems excessive – it should be reduced in length and sharpened to stay a little more focused on the actual data (there seem to be many extrapolations well beyond the scope of the data). Lines 463-465 are super-clear – they encapsulate key findings and could be used as a springboard for the discussion.

We have now shortened the Discussion by removing less critical extrapolations and modified the text to further highlight our key findings.

9. There are many beautifully executed experiments in this paper but I was hoping for an old-fashioned one that might indicate the disruption of the sterol/SM gradient – this would be the Brown-Rose experiment which monitored the trafficking of a GPI-anchored protein through the secretory pathway, showing that it became insoluble in cold TX-100 when it reached the Golgi apparatus. While the molecular basis of this phenomenon can be debated the experiment provides a clear readout of the lipid translation that the authors write about in the Introduction and Discussion. It's not required to do this, but the authors may want to consider if it would help develop their description of the phenotypes that they are studying.

We thank the reviewer for this valuable suggestion, which we will take into account in our future efforts to investigate the precise impact of a disrupted SM gradient on the physicochemical and functional properties of the secretory pathway.

10. If I am not mistaken, there are old data by Wieland and van Meer describing changes in transbilayer asymmetry of SM and GlcCer on BFA treatment – these data essentially report on the phenomenon of the symmetrization of the transbilayer distribution of SM at the ER. This work could be cited/discussed.

We assume the reviewer is referring to the paper by Halter, Neumann, … van Meer (2007, JCB 179, 101-115; www.jcb.org/cgi/doi/10.1083/jcb.200704091) in which the authors used BFA to demonstrate that a portion of GlcCer produced on the cytosolic surface of *cis*Golgi cisternae travels back to the ER to gain access to the glycolipid-producing enzymes in the Golgi lumen. However, this work focuses entirely on GlcCer and does not describe any changes in SM asymmetry upon BFA treatment.

Reviewer #3 (Recommendations for the authors):The central claim – that SM production in the ER affects bilayer asymmetry in the PM – rests mainly on two observations: First, the lipidome analysis of PM isolates shows only a modest decrease in total SM levels in the disease variant, while the Eqt-staining of SM at the outer leaflet shows much reduced SM levels, leading the authors to conclude that SM is also present on the inner leaflet of the PM. Direct evidence of this would be strongly desired but was however complicated by the fact that the cytosolic EqtSM reporter could not detect SM at the inside of the PM. Likewise, the cytosolic EqtSM reporter did not directly detect SM on the cytosolic leaflet of the ER but instead localized to some yet unidentified punctate structures. As the nature of the punctae could not be clarified, an artefact induced by Eqt-SM binding (aggregates? condensates?) cannot be ruled out. Similar concerns apply to the use of EqtSMss in the ER lumen, which under isotonic show strange punctae (most of which cluster around the perinuclear region) that do not convincingly co-localize with calnexin. A clear binding of EqtSMss to calnexin-labeled membranes could only be shown in hypotonic conditions. In summary, while the use of biosensors for studying lipid localization is essential to this study, as the integrity of the cells must be preserved to properly study questions of lipid asymmetry, their use is accompanied by drawbacks that were not fully addressed. In particular, the reason why cytosolic EqtSM is not able to detect SM on the inner leaflet of the PM remains unclear.

We thank the reviewer for raising these critical points, which led us to perform additional experiments and modify the manuscript as outlined below.

As requested by Reviewer #1 (comment #a), we now also used lysenin (LysSM) as alternative to the cytosolic EqtSM reporter. We find that pathogenic SMS2 variants trigger translocation of LysSM to cytosolic puncta that can be labeled by both reporters (new Figure 6B and C). Moreover, new co-localization studies revealed substantial overlap between the cytosolic puncta and EqtSM_SS_-positive puncta that form in the ER lumen of cells expressing pathogenic SMS2 variants (new Figure 6D). Upon hypotonic swelling, the luminal puncta become largely continuous with the ER membrane (Figure 5B) and tend to segregate from the cytosolic puncta (new Figure 6D, bottom row). Under these conditions, the cytosolic puncta give rise to small vesicular structures, implying the presence of a membranous core. As requested by Reviewer #2 (comment #6), we now included a cartoon to illustrate these findings (new Figure 6E). Collectively, our data indicate that cytosolic puncta originate at least in part from the ER and that their formation is not an artefact inherent to the use of EqtSM. They also reinforce the notion that SM produced by pathogenic SMS2 variants is scrambled across the ER bilayer, in line with previous literature on the substrate profiles of ER-resident scramblases (Buton et al., 2002; Chalat et al., 2012).

A puzzling aspect of the above scenario is that the EqtSM_cyto_-positive puncta are largely segregated from ER membrane markers (VAPA, calnexin; Figure 6A and Appendix – figure 5). Whether the condensation of EqtSM into puncta reflects binding of the reporter to pre-existing SM microdomains or involves a clustering of SM-bound reporters driven by a residual pore-forming activity remains to be established. Resolving this issue will require a substantial body of additional work. As outlined in our response to comment #1 of Reviewer #1 and on p. 9 of the manuscript (lines 309-317), we envision that binding of EqtSM to the membrane may alter its curvature and even pinch off SM-enriched membrane regions, analogous to the shedding of extracellular vesicles observed when cells are exposed to the cholesterol/SM binding protein ostreolysin A (Skocaj et al., 2016; https://doi.org/10.1016/ j.bbamem.2016.08.015). In fact, such shedding may help explain why EqtSM_cyto_-positive puncta do not colocalize with a wide range of organellar markers (Appendix – figure 5) and why externally added recombinant EqtSM preferentially stains the positively-curved edges of adherent cells (Figure 7A). While our data underline the value of EqtSM and LysSM as tools for probing SM topology in intact cells, we now added a note of caution regarding their potential impact on the structural organization of SM-rich membranes (p. 9, lines 317-320).

To directly assess whether SM production and scrambling in the ER affect SM asymmetry across the PM and cause a rise in SM levels in the inner PM leaflet, we now also analyzed the distribution of cytosolic EqtSM in cells expressing pathogenic SMS2 variants after treatment with DMSO, an agent that triggers PM vesiculation or “blebbing”. In DMSO-treated U2OS cells expressing the pathogenic variant SMS2^M64R^, we observed numerous small EqtSM_cyto_-positive puncta that are continuous with the PM (see Author response image 2, with PM-associated puncta marked by white arrows). Such puncta were absent in control cells. However, PM staining with the cytosolic EqtSM reporter was modest, also after treatment with the neutral SMase inhibitor GW4869. We suspect that the bulk of EqtSM_cyto_-positive puncta in cells expressing pathogenic SMS2 variants originate from the ER. This may not be surprising, as ongoing SM production and scrambling in the ER would render this organelle into a sink for cytosolic SM reporters. Circumventing this problem is not trivial and would require access to potent SMS inhibitors that are currently not available. While our findings with DMSO-treated cells provide additional support for the notion that SM production in the ER affects its transbilayer distribution at the PM, we acknowledge that the PM staining observed is not very robust. Therefore, we decided not to include these novel data in the revised manuscript.

The connection of SM mislocalization to subcellular cholesterol pools is very interesting, but not fully explored. Somewhat surprisingly, the authors found that ER cholesterol remains low even in conditions of high ER SM levels, and similarly, overall PM cholesterol levels are not affected in the disease model. Yet, detection of Chol at the inner leaflet of the PM by D4H is no longer possible when using SMS2-mutants. Instead, D4H localizes to endolysosomes. The authors do not comment on how this alternative localization potentially arises (increased Golgi-to-endosome trafficking? Cholesterol export defects of the lysosome in conditions of SM abundance?) nor confirm it with alternative methods (such as Filipin staining). Instead, they claim that the previously hypothesized SM pool in the inner leaflet of the PM renders the cholesterol inaccessible to the reporter. Here, again, potential artefacts of the reporter cannot be excluded and should be controlled for. For instance, investigating the cholesterol levels on the outer leaflet using recombinant reporters (as performed for SM) would help clarify if cholesterol distribution itself or indeed just their complex formation with SM is affected.

We agree with the reviewer that more work will be necessary to fully elucidate the impact of SM mislocalization on subcellular cholesterol pools. Whether redistribution of the sterol reporter D4H from the PM to endolysosomes in cells expressing pathogenic SMS2 variants is caused by an enhanced Golgi-to-endosome trafficking of cholesterol, impaired cholesterol export from lysosomes, or changes in the endolysosomal pool of sequestered and free cholesterol remains to be established. Addressing this question will require a quantitative MS-based assessment of cholesterol levels in purified endolysosomes combined with the application of reporters that are able to distinguish sequestered from free cholesterol. These studies are not trivial and, in our opinion, merit a study in its own right.

For better comprehension, the authors could provide a graphical representation of the downstream effects of SM production at the ER and its effects on cholesterolAt several points in the manuscript, the authors use unpublished data to support their arguments. These data should be provided.

As requested, we now included a graphical representation of how pathogenic SMS2 variants affect the subcellular SM distribution, promote translocation of EqtSM_cyto_ to cytosolic puncta, and influence cholesterol organization at the PM (Figure 9E).